

# Enhancing Mobile Aerosol Monitoring with CE376 Dual-Wavelength Depolarization Lidar

Maria Fernanda Sanchez Barrero[1,2], Ioana Elisabeta Popovici[1,2], Philippe Goloub[1], Stephane Victori[2], Qiaoyun Hu[1], Benjamin Torres[1], Thierry Podvin[1], Luc Blarel[1], Gaël Dubois[1], Fabrice Ducos[1], Eric Bourrianne[1], Aliaksandr Lapionak[1], Lelia Proniewski[2], Brent Holben[3], David Matthew Giles[3,4], Anthony LaRosa[3,4]

[1] UMR8518—LOA—Laboratoire d'Optique Atmosphérique, Centre National de la Recherche Scientifique (CNRS), University of Lille, 59000 Lille, France

[2] R&D Department, Cimel Electronique, 75011 Paris, France

[3] Goddard Space Flight Center-NASA, Greenbelt, 20771 MD, USA

[4] Science Systems and Applications, Inc., Lanham, 20706 MD, USA

*Correspondence to*: Maria F. Sanchez Barrero (mariafernanda.sanchezbarrero@univ-lille.fr)

**Abstract.** We present the capabilities of a compact dual-wavelength depolarization lidar to assess the spatio-temporal variations in aerosol properties aboard moving vectors. Our approach involves coupling the lightweight CIMEL CE376 lidar, which provides measurements at 532 nm and 808 nm and depolarization at 532 nm, with a photometer to monitor aerosol properties. The assessments, both algorithmic and instrumental, were conducted at ATOLL (ATmospheric Observatory of liLLe) platform operated by the Laboratoire d'Optique Atmosphérique (LOA), in Lille France. An early version of the CE376 lidar co-located with the CE318-T photometer and with a multi-wavelength Raman lidar were considered for comparisons and validation. We developed a modified Klett inversion method for simultaneous two-wavelength elastic lidar and photometer measurements. Using this setup, we characterized aerosols during two distinct events of Saharan dust and dust smoke aerosols transported over Lille in spring 2021 and summer 2022. For validation purposes, comparisons against the Raman lidar were performed, demonstrating good agreement in aerosols properties with relative differences of up to 12 % in the depolarization measurements. Moreover, a first dataset of CE376 lidar and photometer performing on-road measurements was obtained during the FIREX-AQ (Fire Influence on Regional to Global Environments and Air Quality) field campaign, deployed in summer 2019 over the Northwestern USA. By lidar and photometer mapping in 3D, we investigated the transport of released smoke from active fire spots at William Flats (North East WA, USA). Despite the extreme environmental conditions, our study enabled the investigation of aerosol optical properties near the fire source, distinguishing the influence of diffuse, convective, and residual smoke. Backscatter, extinction profiles, and column-integrated lidar ratios at 532 and 808 nm were retrieved for a quality-assured dataset. Additionally, Extinction Angstrom Exponent (EAE), Color Ratio (CR), Attenuated Color Ratio (ACR) and Particle Linear Depolarization Ratio (PLDR) were derived. In this study, we discuss the capabilities (and limitations) of the CE376 lidar in bridging observational gaps in aerosol monitoring, providing valuable insights for future research in this field.

## 1 Introduction

Improving the knowledge of the spatio-temporal distribution of aerosols and their local, regional and global impact, as well as reducing the uncertainties on aerosols properties is fundamental to quantify their radiative impacts (Boucher et al., 2013).



Following the aerosols transport from the main sources and the evaluation of their complex distribution are therefore needed. Negative effects on human health and economy are accounted to aerosols as well, increasing the demand for continuous air quality control to develop early warning systems, as more frequent and extreme environmental events are detected (Seneviratne et al., 2021). Photometer and lidar are convenient tools to assess the aerosols properties and their impact on climate. To this end, the development of networks plays a key role for aerosols monitoring. Some examples are: the AERONET network (AErosol RObotic

NETwork; Holben et al., 1998) for photometers, EARLINET (European Aerosol Research LIdar Network; Sicard et al., 2015), now part of ACTRIS-ERIC (Aerosol, Clouds and Trace gases Research InfraStructure, European Research Infrastructure Consortium), for Raman lidars and MPLNet (Micro-Pulse Lidar Network; Welton et al., 2001) for micropulse lidars. Studies conducted with multiple network sites allowed to assess the variability of aerosols properties at a regional level, like for dust outbreaks (Ansmann et al., 2003; Papayannis et al., 2008; López-Cayuela et al., 2023) or long-range transport of biomass burning

episodes (Nicolae et al., 2013; Adam et al., 2020). However, laboratories in fixed sites are restricted by their local conditions and position with respect to the aerosol sources. Furthermore, some regions of difficult access, such as oceans or mountains, remain unexplored. Thus, the deployment of mobile laboratories (aboard ship cruises, airplanes or car) provided a solution to fill these observational gaps in networks (Smirnov et al., 2009; Tesche et al., 2009; Müller et al., 2014; Bohlmann et al., 2018; Popovici et al., 2018; Yin et al., 2019).

In recent years, the multispectral sun/sky/lunar CIMEL CE318-T photometer (Barreto et al., 2016), widely used in AERONET sites and designed by CIMEL company, has been fully adapted for automatic sun/lunar tracking during movement on-board ships (Yin et al., 2019). The ship-borne CE318-T photometer is operational and continuously providing Aerosol Optical Depth (AOD) data since January 2021 aboard the Marion Dufresne research vessel, in the frame of MAP-IO (Marion-Dufresne Atmospheric Program Indian Ocean). Likewise, PLASMA (Photomètre Léger Aéroporté pour la Surveillance des Masses d´Air; Karol et al.,

2013) photometer was developed exclusively to track the sun in movement, and has been deployed aboard aircrafts and vehicles during field campaigns (Popovici, 2018; Popovici et al., 2018, 2022; Hu et al., 2019; Mascaut et al., 2022). The ship-borne CE318-T and PLASMA photometers have been adapted and developed respectively in the frame of AGORA-Lab, a common LOA/CIMEL laboratory (https://www.agora-lab.fr/, last access: 24 October 2023).

    Lidar systems are mostly big, complex, require large space, regular maintenance and controlled operational conditions. So,

upgrades for mobile applications are frequently linked to instrumental modifications and/or creation of adapted laboratory platforms. Studies conducted with lidars aboard mobile vectors showed the possibilities to support satellite-based observations (Burton et al., 2013; Warneke et al., 2023), air quality assessment in urban-rural transitions and complex topographies (Royer et al., 2011; Pal et al., 2012; Dieudonné et al., 2015; Shang et al., 2018; Popovici, 2018; Popovici et al., 2022; Chazette and Totems, 2023). Hence, a description of a compact and light mobile system, which integrated a lidar and a sun photometer was first presented

by Popovici et al. (2018). This unique system deployed by LOA, included the CIMEL CE370 mono-wavelength elastic lidar and the PLASMA sun-photometer. For several field campaigns the integrated system performed on-road mobile measurements (Popovici et al., 2018, 2022), showing the versatility of such system for aerosols characterization. On that account, we propose the newest model of CIMEL lightweight lidar, the CE376 dual-wavelength lidar, to enhance aerosols properties.

    The CE376 lidar measures attenuated backscatter profiles at 532 nm and 808 nm and depolarization at 532 nm. Algorithmic and

instrumental assessment took place at ATOLL platform. METIS, an early version of the CE376 lidar, has been continuously performing observations since 2019. In addition, METIS is co-located with a CE318-T photometer and with a high-power multi-wavelength Raman lidar, LILAS, part of ACTRIS-ERIC, which are also considered for comparison and validation. Multiple studies



performed on simultaneous 2-wavelength lidar measurements proposed inversion schemes by establishing a constant ratio between
wavelengths, and/or requiring the aerosols extinction-to-backscatter ratios, i.e. Lidar Ratio (LR), to be known a priori and constant
(Potter, 1987; Ackermann, 1997, 1999; Kunz, 1999; Vaughan, 2004; Lu et al., 2011). Therefore, we propose an inversion scheme
with a 2-wavelength modified Klett inversion, using AOD and EAE from the photometer to constrain the retrievals. Both forms of
Klett solution, backward and forward integration (Weitkamp, 2005), are used according to detection limits at each wavelength.
Profiles of EAE, CR and PLDR are derived later on. In addition, the attenuated total backscatter and ACR are derived directly
from the measurements. Moreover, the aerosols retrievals are validated through comparison with LILAS Raman lidar and we
establish the reliability of our results. Our study not only outlines the findings but discusses the limitations and future implications
of our approach.

A first dataset of co-located CE376 lidar and photometer mobile observations has been obtained during the FIREX-AQ field
campaign, organized over the Northwestern US in summer 2019 (FIREX-AQ white paper, 2019). This campaign, led by NASA
and NOAA, focused on investigating the chemistry and transport of smoke from wildfires and agricultural burning, in addition to
the multiple in-situ instruments deployed in fixed platforms around the region and aboard aircrafts (Warneke et al., 2023). Remote
sensing instruments were installed in both stationary and mobile DRAGON payloads (Distributed Regional Aerosol Gridded
Observations Networks; Holben et al., 2018). Thus, two mobile platforms (2 SUVs) called DMU-1 and DMU-2 (Dragon Mobile
Unit) were equipped with lidars and photometers. The dual wavelength CE376 lidar and ship-borne CE318-T photometer were
installed aboard DMU-1, and the mono-wavelength CE370 lidar and PLASMA photometer on board DMU-2. Both DMUs
performed on-road mobile observations around major fire sources and were able to follow the smoke plumes. Height-resolved
optical properties of fresh smoke aerosols close to active fire sources were retrieved, despite extreme environmental conditions
which limited the performance of the instruments. Hence, in this work we present aerosols properties mapping of selected case
studies during the William Flats fire in Washington State. Both DMU-1 and DMU-2 are considered for the analysis. Notably, our
study provides 3D mapping and temporal evolution of aerosol properties, showcasing the relevance of coupling the CE376 lidar
and CE318-T photometer even under extreme environmental conditions.

The main objective of this work is to show the capabilities of a compact dual-wavelength depolarization lidar to assess the spatio-
temporal distribution of aerosol properties, particularly when it is aboard moving vectors and co-located with a photometer. Thus,
we explore both capabilities and limitations of CE376 in detail, demonstrating how our study contributes to filling observational
gaps within aerosols monitoring networks. This manuscript is distributed as follows. The description of the instruments used is
presented in Sect. 2. An extensive description of the methodology applied to retrieve aerosols properties, using the 2-wavelength
depolarization lidar and photometer is presented in Sect. 3. The results are divided in two parts, Sect. 4 shows the outcomes of the
algorithmic and instrumental assessments that took place at Lille-France. We present 2 case studies for events of dust and dust-
smoke transported over Lille, and the validation of aerosols retrievals with comparisons against a Raman lidar. Section 5 shows
3D mapping and temporal evolution of aerosols properties using the dual-wavelength CE376 lidar and the CE318-T photometer
mobile observations for the first time. Case studies from the FIREX-AQ campaign presenting the optical properties of fresh smoke
aerosols close to the source are presented. Finally, Sect. 6 summarizes the results and presents the conclusions and perspectives of
this work. The instrumental, algorithmic limitations and the uncertainties are discussed throughout the sections.



**2 Remote sensing instrumentation**

This section is dedicated to the description of the remote sensing instruments used in this study, all of them able to perform
measurements during movement. Section 2.1 presents the new CIMEL CE376 lidar with up to two wavelengths and depolarization channels. Section 2.2 describes the two photometers that were integrated to mobile systems to retrieve aerosols optical properties.

**2.1 Lidars**

The **CE370 lidar** is an eye-safe micro-pulse lidar (Pelon et al., 2008; Mortier et al., 2013; Popovici et al., 2018) operating at 532 nm with 20 µJ pulse energy at 4.7 kHz repetition rate (table 1). It is designed with a shared transmitter-receiver telescope connected
through a 10 m optical fiber to the control and acquisition system. The backscattered signal is detected by photon counting with an avalanche photodiode (APD). The CE370 lidar was designed by CIMEL Electronique to monitor aerosols and clouds properties up to 15-20 km with a vertical resolution of 15 m. For several field campaigns, the CE370 lidar embarked on mobile platforms has demonstrated the viability to characterize vertical aerosols properties in movement (Popovici et al., 2018, 2022). Therefore, the latest lidar model, CE376, operable up to two wavelengths is proposed to replace the CE370 lidar and continue the developments
towards mobile aerosols monitoring (https://www.cimel.fr/solutions/ce376/, last access: 24 October 2023). The CE376 lidar is designed to support up to 2 wavelengths and depolarization measurements within different model configurations (G, GP, GPN, N). In this study we use the CE376 GPN (Green Polarized Near-infrared) model that is described as follows.

The **CE376 GPN lidar** is an autonomous, lightweight and compact micro-pulse lidar. The lidar operates at 2 wavelengths, 532 nm and 808 nm, with 5-10 µJ and 3-5 µJ pulse energy, respectively, at repetition rate of 4.7 kHz (table 1). Measurements of elastic
backscattered light at both wavelengths and depolarization at 532 nm are acquired. For both systems used in this work (METIS and FIREX-AQ), the laser source at 532 nm has been replaced with one of higher pulse energy (not eye safe) to increase the Signal to Noise Ratio (SNR). The Emission-Reception design consists of two Galilean telescopes in biaxial configuration. The simplified 2D layout of the lidar system is presented in Fig. 1. Light pulses at 532 nm from a frequency doubled Nd:YAG laser source are transmitted through an arrangement of dichroic mirrors and collimation lenses on the green emission system. Same, a simplified
optical system including a pulsed narrow bandpass laser diode source, optical fiber and collimation lenses emits light pulses in the near infrared (NIR) at 808 nm. The elastic backscattered light is collected, collimated and filtered in the reception at each emitted wavelength, and detected with APDs in photon counting mode. Electronic cards developed by CIMEL communicate with the control and acquisition software.

Linear depolarization measurements at 532 nm are also acquired by separation in parallel (co-polarized) and perpendicular (cross-
polarized) components of the backscattered light using a Polarizing Beamsplitter cube (PBS) in the reception. The PBS is a Thorlabs CCM1-PBS25-532 with reflectivities Rp and Rs and transmittances Tp and Ts (subscripts p and s for parallel and perpendicular polarized light with respect to the PBS incident plane). Typical values on commercial cubes correspond to Rs>Rp with Rs~1, Tp>Ts and considering Rp=1-Tp and Rs=1-Ts, i.e., higher reflectivity for the perpendicular incident polarized light, and higher transmittance for the parallel component (Freudenthaler et al., 2009). A manual Half-Wave Plate (HWP) in front the
PBS controls the polarization angle of the incident light with a precision of 2 degrees. Measured signals behind the PBS on the reflected and transmitted branches are named parallel (//) or perpendicular (⊥) according to the reception configuration. More details on the depolarization measurements can be found in Sect. 3.1.1.



For mobile applications, the CE376 lidar is coupled with a GPS module to derive the exact position during measurements. The integration of the geolocation and lidar observations are accounted on the data pre-processing, described in Sect. 3.1.2.

**145   2.2 Photometers**

The CIMEL CE318-T photometer has been adapted for mobile applications. The PLASMA photometer has been developed exclusively for mobile observations. Both instruments follow and meet the AERONET standards and are included in the data processing line. Therefore, automatic near real time aerosols properties can be retrieved (https://aeronet.gsfc.nasa.gov, last access: 23 October 2023), without cloud screening at data level 1.0 and with cloud screening at data level 1.5. After calibration, quality
assured data at level 2.0 is also acquired (Smirnov et al., 2000). Both photometers are used in this work and are briefly described below.

The **sun/sky/lunar CIMEL CE318-T photometer** developed by Cimel Electronique (Barreto et al., 2016) performs both daytime and night-time observations. Direct solar/lunar measurements are collected automatically through 9 channels (340, 380, 440, 500, 670, 870, 936, 1020 and 1640 nm) deriving spectral AOD, with accuracy of 0.01. EAE is determined by pairs of AOD values at
different wavelengths, providing information on the size distribution of aerosols ( Kusmierczyk-Michulec, 2002). Moreover, multi-angular sky radiance measurements are acquired in the almucantar plane during daytime. Aerosols microphysical properties, such as Volume Size Distribution (VSD), complex refractive index and single-scattering albedo can be also retrieved through inversion procedures (Dubovik and King, 2000). In the last few years, the photometer has been adapted for mobile measurements aboard cruise ships to cover oceans. The ship-borne CE318-T described by Yin et al. (2019) and developed at LOA, in the frame of
AGORA-Lab, enables AOD acquisition during movement. The system is coupled with a compass and GPS modules, obtaining information on date, time, geolocation, heading, pitch and roll to target the sun/moon continuously. With the help of an accelerated tracking feedback loop, the system switches into its regular tracking mode to improve measurements quality. Downward sky radiances are also measured with additional information (from GPS and compass) for each almucantar angle to have accurate knowledge of the observation geometry. The ship-borne CE318-T is operational and continuously measuring since January 2021
on board Marion Dufresne research vessel, as part of MAP-IO project (http://www.mapio.re, last access: 9 October 2023). Likewise, a second instrument with upgraded software has been installed and it is performing measurements since April 2023 aboard Marion Dufresne vessel. In this manuscript we will show the integration of the CE318-T photometer and CE376 lidar with measurements at fixed location (Sect. 4) and for the first time on-board a car during FIREX-AQ campaign (Sect. 5).

The **sun-tracking-photometer PLASMA** developed by LOA and SNO/PHOTONS has the capability of performing direct solar
radiation measurements during movement. The instrument is easy to set up and transport due to its light and compact design (~5 kg and 23 cm height). PLASMA has 9 spectral channels at 339, 379, 440, 500, 674, 870, 1019 and 1643 nm and 937 nm for water vapor measurements. Spectral AOD with accuracy of 0.01 and EAE are derived from the direct solar radiation measurements (Karol et al., 2013). A more detailed description of the instrument and its application to airborne measurements are presented by Karol et al. (2013). PLASMA on-board an aircraft during AEROMARINE field campaign at Reunion island (Mascaut et al., 2022)
shows the alternative use of the instrument to obtain AOD and EAE vertical profiles during the aircraft's ascendent/descendent trajectories. The integration of PLASMA and CE370 lidar performing on-road mobile measurements (Popovici et al., 2018, 2022; Hu et al., 2019) has been carried out during several campaigns. Likewise, PLASMA and CE370 lidar were coupled to perform mobile measurements during FIREX-AQ campaign (Sect. 5).



**3 Methodology**

In this manuscript, we describe extensively the methodology applied to retrieve aerosol optical properties from measurements of the CE376 GPN lidar, named simply as CE376 hereafter. Detailed description on methods and corrections applied to the mono-wavelength CE370 lidar can be found in previous works (Pelon et al., 2008; Mortier et al., 2013; Popovici et al., 2018). For this study, two early versions of CE376 are used, one performing continuous observations at Lille, France and the other installed on-board a mobile platform during FIREX-AQ field campaign. Data treatment and quality assurance for both types of measurements,

fixed location and on-board mobile platform, follow the same steps with exceptions mainly on the determination of molecular contributions.

In this section, details from pre-processing to aerosol optical properties retrievals are presented. Section 3.1 describes the atmospheric parameters derived directly from the observations. The Volume Linear Depolarization Ratio (VLDR) is described in Sect. 3.1.1. The total attenuated backscatter is described in Sect. 3.1.2 and the ACR definition is presented in Sect. 3.1.3. Section

3.2 presents the inversion methods applied to obtain aerosol optical properties. The methodology described below is summarized with a block diagram in Fig. 2, showing the atmospheric optical properties derived from the CE376 and CE318-T measurements.

**3.1 Lidar data-processing**

The light backscattered by molecules and aerosols at a distance *r* from the lidar, is collected by a telescope and detected by photon counting with an APD. Considering the lidar equation (Vladimir A. Kovalev and William E. Eichinger, 2004; Weitkamp, 2005),

the detected elastic backscattered signal can be described as Eq. (1).

$$RCS(\lambda, r) = C_{L,\lambda}[\beta_m(\lambda, r) + \beta_a(\lambda, r)]T_m^2(\lambda, r)T_a^2(\lambda, r) \tag{1}$$

$$T_m^2(\lambda, r)T_a^2(\lambda, r) = \exp\left(-2\int_0^r \alpha_m(\lambda, r')dr'\right)\exp\left(-2\int_0^r \alpha_a(\lambda, r')dr'\right) \tag{2}$$

The Range Corrected Signal (RCS) [Ph s⁻¹ m²] is the detected signal after background, range dependence (*r²*) and overlap *O(r)* corrections. RCS profiles are obtained for each detection channel of the CE376, i.e., for co (parallel) and cross (perpendicular)

polarized signals at 532 nm, $RCS(532//, r)$ and $RCS(532 \perp, r)$ respectively, and total signal at 808 nm, $RCS(808, r)$. The right side of Eq. (1) is therefore described only in terms of atmospheric optical properties correlated to the measured signal RCS through a calibration constant $C_{L,\lambda}$ in [Ph s⁻¹ m³ sr]. The term $\beta(r)$ is the backscatter coefficient [m⁻¹sr⁻¹]. $T^2(\lambda, r)$ is the non-dimensional two-way atmospheric transmittance defined in Eq. (2), where *α(r)* is the extinction coefficient [m⁻¹]. Subscripts *m* and *a* represent contributions of molecules and aerosols, respectively. Background noise and overlap corrections at each detection channel are

applied in the same way as for CE370 lidar and are described in previous works (Pelon et al., 2008; Mortier et al., 2013; Popovici et al., 2018).

The integral $\int_0^r \alpha_a(\lambda, r')dr'$ in Eq. 2 is also known as AOD, and it is directly measured by photometer for the total atmospheric column. Therefore, hereafter subscripts *ph* and *lid* will be used to differentiate optical properties from photometer and lidar, respectively. The AOD_ph for the lidar wavelengths, 532 nm and 808 nm, are interpolated by following the Ångström law using

AOD_ph at 440 nm and EAE_ph(440/880).



The main sources of uncertainties on the RCS profiles come from the overlap correction in the lower troposphere, and from the background irradiance in the higher atmosphere (Sassen and Dodd, 1982; Welton and Campbell, 2002; Guerrero-Rascado et al., 2010; Popovici et al., 2018; Sicard et al., 2020). For RCS at 532 nm from both CE376 systems used in this work, considerable underestimations on the incomplete overlap region (< 2.5 km) are observed for temperatures below 17 °C and above 35 °C, adding

error into the lower range of the profiles. The profiles $RCS(532 \perp, r)$ and $RCS(808, r)$ are the most affected by the solar background, reducing the detection limits by day. The relative error induced by the APD in photon counting mode is less than 5%.

For mobile observations, a GPS module is coupled to the CE376 lidar. The geolocation is measured with high temporal resolution (1 s). For each RCS profile, we determine its latitude, longitude and altitude above sea level (asl) by comparing recorded times for both GPS and lidar. We derive the velocity of the mobile platform from the geo-location and time to flag the stationary and mobile

measurements for further analysis. In Sect. 5, case studies of mobile observations within a complex topography are presented. Thus, we took special attention on pairing geo-location and RCS profiles to assess properly the complexity of the terrain.

### 3.1.1 Volume Linear Depolarization Ratio

The total RCS and VLDR at 532 nm defined below in Eq. (3) and Eq. (4-5) respectively are derived following the methods described by Freudenthaler et al. (2009). The signals measured by the detectors behind the PBS are $RCS_R$ on the reflected branch

and $RCS_T$ on the transmitted branch. Rotating the HWP, the angle φ between the plane of polarization of the laser and the incident plane of the PBS can be changed for two arrangements (φ=0º or 90º). For commercial PBS cubes (Rs>Rp and Tp>Ts), the system configuration at φ=0º is defined when the parallel polarized signal is measured in the transmitted branch of the PBS. Therefore, $RCS_T(r)$ = RCS(532//, $r$) , $RCS_R(r)$ = $RCS(532 \perp, r)$, the measured signal ratio δ*(r)=RCS(532 $\perp$,$r$)/RCS(532//,$r$) and the VLDR defined as Eq. (4). Moreover, to reduce noise and errors from cross-talk effects, the configuration φ=90º can be also

considered, with $RCS_T(r)$ = $RCS(532 \perp, r)$, $RCS_R(r)$ = RCS(532//,$r$), δ*(r)=RCS(532//,$r$)/RCS(532 $\perp$,$r$) and the VLDR defined as Eq. 5. The relative amplification factor V* is obtained from calibration.

$$RCS(r) = RCS_R(r) + V^* RCS_T(r) \qquad (3)$$

$$\delta^v(r) = \left[ R_p - \frac{\delta^*(r)}{V^*} T_p \right] \Big/ \left[ \frac{\delta^*(r)}{V^*} T_s - R_s \right] \qquad \text{for} \qquad \text{φ=0º} \qquad (4)$$

$$\delta^v(r) = \left[ \frac{\delta^*(r)}{V^*} T_p - R_p \right] \Big/ \left[ R_s - \frac{\delta^*(r)}{V^*} T_s \right] \qquad \text{for} \qquad \text{φ=90º} \qquad (5)$$

Under cloud free and stable atmospheric conditions, V* calibration coefficient is calculated using the ±45° calibration (Freudenthaler et al., 2009). The HWP rotates the angle of the incident polarization plane φ by means of 2θ with θ precision of 2°. The error induced by the uncertainty in φ represent less than 5% of error on V* for VLDR values up to 0.3 (Figure 2, Freudenthaler et al., 2009). Moreover, to improve depolarization measurements, wire-grid polarizers were added to the PBS to reduce the cross-talk. However, additional errors during the calibration and in regular measurements can come from polarizing optical components

that need detailed characterization (Freudenthaler, 2016), which are not considered in this work. For future versions of the CE376, a motorized PBS mount will be integrated.



### 3.1.2 Total Attenuated Backscatter

For quality assurance of lidar profiles, we follow the standard Rayleigh fit procedure (Freudenthaler et al., 2018), meaning that we normalize $RCS(\lambda, r)$ to the molecular profile $\beta_m(\lambda, r)T_m^2(\lambda, r)$ at a distance $r_{ref}$ where we assume a free aerosols zone, i.e. $\beta_a(\lambda, r_{ref}) = 0$. The molecular backscatter coefficients $\beta_m(\lambda, r)$ and the two-way molecular transmittance $T_m^2(\lambda, r)$ are calculated using the pressure and temperature profiles from standard atmosphere models or from available radiosonde data. This method is recurrently applied to signals from each channel of the CE376, especially during night time when SNR is higher. Moreover, we use the same considerations to determine the calibration constant $C_{L,\lambda}$, for total signals $RCS(532, r)$ and $RCS(808, r)$. Hence Eq. (6) can be derived from Eq. (1).

$$C_{L,\lambda} = RCS(\lambda, r_{ref}) / [\beta_m(\lambda, r_{ref})T_m^2(\lambda, r_{ref})T_a^2(\lambda, r_{ref})] \tag{6}$$

The aerosols transmittance term $T_a^2(\lambda, r_{ref})$ can be calculated if $AOD_{ph}$ is available. Assuming that no aerosols are present above $r_{ref}$ we have $T_a^2(\lambda, r_{ref}) = \exp(-2\,AOD_{ph}(\lambda))$. If there are no changes on the lidar system configuration, the $C_{L,\lambda}$ stability over time is mainly controlled by the laser energy and the opto-mechanical stability. Then the total attenuated backscatter $\beta_{att}(\lambda, r)$ is defined by Eq. (7).

$$\beta_{att}(\lambda, r) = RCS(\lambda, r) / C_{L,\lambda} \tag{7}$$

### 3.1.3 Attenuated Color Ratio

The CR, defined as the ratio of aerosol backscatter at two different wavelengths, has been used to discriminate clouds from aerosol layers and eventually for aerosol typing (Omar et al., 2009; Burton et al., 2013; Wang et al., 2020; Qi et al., 2021). In particular, CALIPSO (Cloud Aerosol Lidar and Infrared Pathfinder Satellite Observation) algorithms use the layer mean total attenuated backscatter as a first approximation of the aerosol backscatter $\bar{\beta}_{att} = [1/(r_{top} - r_{base})] \int_{base}^{top} \beta_{att}(r')dr'$ and defines the layer-integrated attenuated color ratio as $\chi' = \bar{\beta}_{att}(1064)/\bar{\beta}_{att}(532)$. Then both layer-integrated features are used for classification of stratospheric aerosols (Vaughan et al., 2004; Omar et al., 2009; Kim et al., 2018). Similarly, the attenuated total backscatter corrected by the two-way molecular transmittance term is considered as a first approximation of the aerosol backscatter. Therefore, the ACR for all the ranges is defined by Eq. (8).

$$ACR(r) = \frac{\beta_{att}(808, r)\,T_m^{-2}(808, r)}{\beta_{att}(532, r)\,T_m^{-2}(532, r)} = \frac{[\beta_m(808, r) + \beta_a(808, r)]}{[\beta_m(532, r) + \beta_a(532, r)]}\exp(-2\int_0^r[\alpha_a(808, r') - \alpha_a(532, r')]dr') \tag{8}$$

The ACR contains information of molecules and aerosols and mostly provides insights on the aerosols size. For purely molecular atmosphere, the ACR is reduced to the ratio of molecular backscatter coefficients and ACR~0.19. Clouds are generally composed of large particles, compared to the lidar wavelengths, so the backscatter and extinction coefficients are not expected to show spectral variation. Therefore, ACR values for clouds are likely to be close to 1. Assuming that only one type of aerosols is present and homogeneously distributed in the atmospheric column, the exponential term goes nearly constant and the ACR is controlled by the ratio $\beta_a(808, r)/\beta_a(532, r)$. Under this rough assumption, ACR values for aerosols are between 0 and 1, with low values for fine aerosols and close to 1 for large particles.



### 3.2 Aerosols Optical Properties

By solving the Eq. (1) and assuming a constant LR, we retrieve $\beta_a(\lambda, r)$ as in Eq. (9) (Weitkamp, 2005), well-known as Klett
solution (Klett, 1985). A constant extinction-to-backscatter ratio of $8\pi/3$ sr for molecules at all wavelengths is considered. For
mobile measurements, we also consider surface altitude asl for each RCS profile to model correctly the molecular profiles,
$\beta_m(\lambda, r)$ and $\alpha_m(\lambda, r)$.

$$\beta_a(\lambda, r) = \frac{RCS(\lambda, r)\exp\left[-2(LR(\lambda)-8\pi/3)\int_{r_b}^{r}\beta_m(\lambda, r')dr'\right]}{\frac{RCS(\lambda, r_b)}{\beta_a(\lambda, r_b)+\beta_m(\lambda, r_b)}-2\,LR(\lambda)\int_{r_b}^{r}RCS(\lambda, r')\exp\left[-2(LR(\lambda)-8\pi/3)\int_{r_b}^{r'}\beta_m(\lambda, r'')dr''\right]dr'} - \beta_m(\lambda, r) \qquad (9)$$

The boundary conditions are given by the position of $r_b$ and therefore two forms of the Klett solution are specified. The far-end
with backward integration given by $r_b = r_{ref}$, is well known as backward (BW) solution and takes the same considerations of
Rayleigh fit (Sect. 3.1.2). It is the most used form of Klett solution, but it has an obvious difficulty when defining $r_{ref}$. The near-
end solution with forward integration or forward (FW) solution is given by $r_b = r_o$, where $r_o$ is close to the ground. Thus, the total
backscatter is $\beta_a(\lambda, r_o) + \beta_m(\lambda, r_o) = \beta_{att}(\lambda, r_o)/T_m^2(\lambda, r_o)T_a^2(\lambda, r_o)$, assuming that aerosol transmittance close to the ground
is roughly 1. Due to the incomplete overlap and lidar's instability, especially for high power and complex systems, the FW
solution is usually not considered. However, it can be applied on measurements from ceilometer-type systems like the 808 nm
channel of CE376, which has available measurements close to the ground and stable configuration. On the other hand, the effective
LR can be derived, for both BW and FW, based on iterative calculation of the solution and constraint by available AOD$_{ph}$ (Mortier
et al., 2013).

During night time measurements, the detection limits (using SNR=1.5 on 30 minutes averaged profiles) for all CE376 channels is
higher than 10 km, so we can usually meet an aerosol free zone ($r_{ref}$) for both 532 nm and 808 nm wavelengths. Therefore, the BW
Klett solution can be applied for both wavelengths. Nevertheless, during daytime, strong solar background light limits the detection
to ~10 km and below 4 km for 532 nm and 808 nm, respectively. Thus, the BW Klett solution for 532 nm can still be applied, but
not for 808 nm. However, the blind zone and complete overlap are below 150 m and ~1 km, respectively, for 808 nm, in contrast
with 400 m and ~2.5 km, respectively for 532 nm. Therefore, we consider FW Klett solution suitable for RCS profiles at 808 nm
during daytime. Taking into account all these considerations, we propose a modified 2-wavelength inversion scheme as follows:

   a) BW Klett solution: applied to RCS total signals and constrained by AOD$_{ph}$ at both wavelengths 532 nm and 808 nm. The
   $r_{ref}$ for each wavelength is searched automatically within a threshold a-priori defined (ex. 6 km to 10 km), and determined
   by minimizing the root mean square error with respect to the molecular signal. We retrieve LR($\lambda$), $\beta_a(\lambda, r)$ and $\alpha_a(\lambda, r)$
   at both wavelengths.
b) FW Klett solution (when $r_{ref}(532) > r_{lim}(808)$): is applied to RCS at 808 nm if the $r_{ref}$ determined for 532 nm is higher
   than the detection limit ($r_{lim}$) for 808 nm. We constrain the solution by an estimated AOD at 808 nm (AOD$_{est}$). AOD$_{est}$,
   defined in Eq. (10), is derived from the lidar retrievals at 532 nm and the interpolated EAE$_{ph}$ for the pair of wavelengths
   532 nm and 808 nm.

$$AOD_{est}(808) = \left[\int_{r_o}^{r_{lim}}\alpha_a(532, r)dr\right]\left(\frac{808}{532}\right)^{-EAE_{ph}} \qquad (10)$$

305   c) Extinction Angstrom Exponent profile ($EAE_{lid}$): is derived from 2 $\alpha_a(\lambda, r)$ and defined as $EAE_{lid}(r) = (-ln[\alpha_a(532, r)/\alpha_a(808, r)])/ln[532/808]$. This parameter gives insights on the vertical distribution of aerosols size,



EAE values close to 0 indicate dominant presence of coarse mode aerosols and values higher than 1 are related to the fine mode aerosols.

    d) Color Ratio (CR): is defined as the ratio between the aerosol backscatter at 808 nm and 532 nm $CR(r) = \beta_a(808, r)/\beta_a(532, r)$ and it is described in Sect. 3.1.3 along with the ACR.

    e) Particle Linear Depolarization Ratio (PLDR): is defined by Eq. (11), where the molecular depolarization ratio $\delta^m$ is the theoretical value according to the bandwidth of the filter in front the half-waveplate in a CE376 system ($\delta^m \sim 0.004$). R= $(\beta_a(r) + \beta_m(r))/\beta_m(r)$ is known as the backscatter ratio and $\delta^v(r)$ is the VLDR profile derived directly from depolarization measurements (Sect. 3.1.1). Furthermore, PLDR gives insights on the vertical distribution of aerosols shape, low values (close to 0) indicate the predominance presence of spherical aerosols. Values above 0.20 correspond to predominant presence of non-spherical aerosols like dust or ice crystals in cirrus clouds.

$$\delta^p(r) = \frac{[1+ \delta^m]\, \delta^v(r)R(r) - [1 + \delta^v(r)]\, \delta^m}{[1+ \delta^m]R(r) - [1 + \delta^v(r)]} \tag{11}$$

A first evaluation of uncertainties at each step in the data processing are approached using first order derivatives. Thus, error propagation guidelines presented in the literature were followed (Russell et al., 1979; Sasano et al., 1985; Kovalev, 1995, 2004; Rocadenbosch et al., 2012; Sicard et al., 2020; Welton and Campbell, 2002; Morille et al., 2007). The main error sources are related to the overlap function estimation, background noise, lidar constant and depolarization calibrations. Therefore, standard deviations from the overlap function and calibrations are considered, and propagated from the RCS and VLDR to the aerosol retrievals. The uncertainty on the LR is roughly estimated by the convergence within the AOD uncertainties (0.01) in the iterative Klett solution. Errors on the molecular optical properties are negligible.

The data processing and inversion scheme presented in this section are the first steps towards near real time observations integrating CE376 lidar and CE318-T photometer. Therefore, the capabilities for continuous monitoring of aerosols properties in fixed and mobile observatories are enhanced and presented through case studies in the following sections.

## 4 Atmospheric Observations at Lille, France

In this section we present the analysis and validation of data from an early version of the CE376 lidar, operational at a fixed location in the metropolitan area of Lille, France. In Sect. 4.1, a description of the site and instruments used for this study are presented. Selected case studies and validation of optical properties derived from the CE376 measurements presented through comparisons with a reference lidar are presented in Sect. 4.2.

### 4.1 ATOLL observatory

METIS is an early version of the CE376, continuously performing at ATOLL at University of Lille (50.61° N, 3.14° E, 60 m asl). The platform is also equipped with online in-situ and other remote sensing instruments providing valuable information on aerosol properties and cloud-aerosol interactions. ATOLL platform is one of the AERONET calibration centers and it is an ACTRIS-ERIC facility. The location is mainly influenced by urban-industrial emissions, marine aerosols (~80 km from the nearest coast), and seasonal pollen outbreaks (Veselovskii et al., 2021). Likewise, events of long-range transport impact the region with aerosols from Saharan mineral dust storms (Veselovskii et al., 2022), North American wildfires (Hu et al., 2019, 2022) and volcanic eruptions (Mortier et al., 2013).



METIS is operational at ATOLL platform since 2019 in the frame of AGORA-Lab. METIS depolarization measurements setup currently follows a configuration with φ=90º, measuring the parallel component on the PBS reflected branch. So $RCS_R(r) = RCS_{532\parallel}(r)$, $RCS_T(r) = RCS_{532\perp}(r)$, $\delta^*(r)=RCS_{532\,//}(r)/RCS_{532\perp}(r)$ and the VLDR vertical profiles are defined by Eq. (5). Wire-grid polarizers behind the PBS branches are used to reduce the cross-talk in the signals (Tp~1, Ts~0 and Rp~0, Rs~1). The

continuous measurements are ensured by setting the lidar in a temperature-controlled room and using a high transmittance glass on the roof. Moreover, METIS is collocated with a CE318-T photometer and with LILAS (LIlle Lidar AtmosphereS) ACTRIS lidar, both considered for this study.

LILAS is a high-power Mie-Raman-Depolarization-Fluorescence lidar developed and upgraded by LOA and CIMEL since 2013. From its simultaneous multiple wavelength measurements, independent height-resolved optical properties are derived: 3

backscatter (355 nm, 532 nm, 1064 nm), 2 extinction (355 nm, 532 nm), 3 particle depolarization ratio (355 nm, 532 nm, 1064 nm) and 1 fluorescence backscatter (at 466 nm) profiles. A detailed description of LILAS system, retrievals and uncertainties can be found in previous works (Bovchaliuk et al., 2016; Hu et al., 2019, 2022; Veselovskii et al., 2022). The aerosol optical properties retrieved with METIS at 532 nm are validated by intercomparisons with LILAS.

Molecular coefficients are modeled using radiosonde measurements from 3 stations near Lille, depending on availability.

Beauvechain (50.78° N, 4.76° E, Belgium) and Herstmonceux (50.90° N, 0.32° E, England) from Wyoming University database (https://weather.uwyo.edu/upperair/sounding.html, last access: 23 October 2023), and Trappes (48.77° N, 1.99° E, France) from Meteo-France database (https://donneespubliques.meteofrance.fr, last access: 23 October 2023). Beauvechain is the closest site, about 120 km away from Lille, Herstmonceux is 200 km and Trappes is 240 km far from Lille.

**4.2 Continuous observations and comparisons with reference lidar**

Since the installation of METIS at ATOLL several studies and instrumental assessments took place in order to improve mainly the depolarization measurements. From first comparisons of METIS and LILAS, an important bias between depolarization measurements were detected (>20 %). The roof glass window was tempered, had an anti-reflective coating and suffered deformations due to its size and weight. All these created biases on the depolarization measurements. Currently, a frame designed to contain four windows is placed instead, avoiding deformations due to glass weight. The glass material was also changed to an

extra-clear glass and the windows are set up on the frame using silicone in order to avoid adding stress to the glass.

In the following case studies, continuous observations of METIS and comparisons with LILAS are presented, with METIS under two different conditions of measurement. The first case is METIS without roof window during an event of Saharan dust transported over Lille in spring 2021. The second case is METIS in the current configuration for continuous measurements during a recent event of dust and smoke transported over Lille in summer 2022.

**4.2.1 *Saharan dust transport over Lille (31 March to 2 April 2021)***

Saharan dust layers transported over Lille are frequently observed and monitored with both METIS and LILAS. One of these events took place from 31 March to 02 April 2021. An overview of the METIS and photometer measurements is presented in Fig. 3. During this event the roof window of METIS was open on 1 April beginning with 07:00 UT, represented by the black dotted line in Fig. 3 panels (a) and (b). The impact of the roof window on the depolarization measurements can be observed, VLDR values



being higher by 0.02 when METIS is with the roof window. For this case, only VLDR values without window are considered for analysis.

The dust event had a period of strong aerosol loading during the night of 31 March 2021 to the afternoon of 1 April 2021. Intrusions of aerosol layers between 1.5 km to 8 km asl were observed, with high VLDR values, on average of 0.20 ±0.04 (1 April 2021, 7:00-19:00 UT), indicating the presence of non-spherical aerosols. $AOD_{ph}$ values at 532 nm and 808 nm increase up to 1 and 0.9,

respectively. $EAE_{ph}$ (532/808) decreases from 1.4 to 0.2 associated to the increase of coarse mode particles concentration. Additionally, VSD derived from photometer observations during 1 April 2021 (Fig. 4) show the strong predominance of aerosols in the coarse mode with an effective radius of 1.7 μm. Thus, with the identified non-spherical coarse particles, the presence of dust is confirmed. Towards the night of 1-2 April 2021, the dust layers slowly vanish, while a peak of pollution develops close to the surface. A shallow boundary layer (<500 m) with a strong inversion at the top constrains the mixing of dust within the boundary

layer. During the day of 2 April the $EAE_{ph}$(532/808) increases up to 1.5 and the VLDR decreases below 0.1.

For comparisons of METIS and LILAS, averaged profiles on 1 April between 20:00 to 22:00 UT were used, when Raman measurements from LILAS were available. Aerosols optical properties were retrieved with the modified 2-wavelength method for METIS CE376 lidar and Raman inversion is used for LILAS. Molecular coefficients were calculated using the radiosonde data taken at 00:00 UT on 2 April 2021 from the station Herstmonceux. Lunar measurements were not acquired until later that night,

so the two closest pair of $AOD_{ph}$ were considered to constrain the inversion for METIS at 1 April 2021 17:50 and 2 April 2021 00:45 UT. Hence, backscatter and extinction profiles at 532 nm and 808 nm for METIS and at 532 nm for LILAS were retrieved and are presented in Fig. 5 panels (a) and (b). VLDR and PLDR at 532 nm for both lidars are also compared (Fig. 5c), as well as LR (Fig. 5f). The ACR and CR of 808-532 nm from METIS are presented (Fig. 5e) as well as EAE (532/808) from METIS and the photometer (Fig. 5d). The first 2 km of the RCS at 532 nm are influenced by relative errors of 5 % at 2 km going towards 20

% at 500 m due to the overlap estimations. In the case of RCS at 808 nm, the influence of overlap error goes from 5 % at 1 km towards 10 % at 150 m. Therefore, to avoid artifacts on the retrievals, RCS values below 500 m are considered constant for both wavelengths. Likewise, PLDR, EAE and CR values are not shown when the aerosol backscatter at 532 nm is less than 0.3 Mm$^{-1}$ sr$^{-1}$ and below 500 m.

Backscatter and extinction profiles comparisons show good agreement between the CIMEL CE376 elastic lidar and LILAS Raman

lidar. The differences in extinction observed are related to the constant LR of 54 ± 3 sr for METIS retrievals at 532 nm. From the profile of LR at 532 nm for LILAS (Fig. 4f), we can see that the first layer between 1.5-3 km asl is 48 sr on average, in contrast with 72 sr for the second layer between 3.3-4.7 km asl. Thus, a better agreement in the lower layer than within the second layer especially for extinction coefficients is observed. From METIS retrievals, the first layer extinction values are in average 61 ± 14 Mm$^{-1}$ and 52 ± 10 Mm$^{-1}$ at 532 nm and 808 nm, respectively. Extinction values in the second layer are in contrast slightly lower,

43 ± 3 Mm$^{-1}$ and 35 ± 6 Mm$^{-1}$ at 532 nm and 808 nm, respectively. The LR at 808 nm resulted from the retrievals is 69 ± 4 sr. Absolute differences up to 0.02 for METIS VLDR profile with respect to LILAS are observed. METIS shows VLDR and PLDR values within the two layers of 0.14 ± 0.02 and 0.36 ±0.05, respectively, comparable to values reported in previous works for Saharan dust transport (Ansmann et al., 2003; Haarig et al., 2022; Floutsi et al., 2023). Lower EAE values for the first layer (0.4 ± 0.1) were observed for the first layer compared to 0.5 ± 0.1 for the second layer. The ACR (808/532) and CR (808/532) profiles

show values of 0.42 ± 0.05 and 0.69 ± 0.14, respectively, for the lower layer and 0.38 ± 0.04 and 0.65 ± 0.12 at the second layer. These results suggest the presence of two different air masses, with larger dust aerosols in the lower layer, which it is also shown in the LR profile from LILAS lidar.



METIS showed VLDR values 10 % higher than LILAS under the same operational conditions. This bias comes from differences on the optical design proper to the instruments and that METIS uses a manual half-wave plate for the polarization calibration while
LILAS uses a motorized PBS mount with an obvious higher precision.

### 4.2.2 Saharan dust and Smoke transport over Lille (17 to 20 July 2022)

Several heatwaves crossed Europe during spring-summer 2022, meaning that air masses from the equatorial region (North Africa) moved northwards pushing temperatures up in several areas, especially in the Western Europe. The unusual long periods of heat since spring intensified the dry conditions during summer. Moreover, due to the vegetation dryness and the extreme high
temperatures, multiple fires were detected in Southwestern Europe in July-August 2022. Unprecedented wildfires have broken out on 12 July 2022 in the Gironde department, Southwestern France, intensified by a heatwave passing with strong winds, over ~270 $km^2$ of burned surface were accounted in the region with the highest forest losses in France. During this event, biomass burning aerosols injected to the atmosphere by the wildfires got mixed with the mineral dust transported within the hot air masses. Therefore, at the time that the heatwave traversed Lille, we detected both dust and smoke in the atmospheric column. For this case,
METIS was performing measurements under the current operational conditions, i.e., adapted roof window and air conditioning. To assess the continuity of the aerosol optical properties, the closest data points are used to constrain the inversion when measurements from photometer are not available.

An overview of the retrieved aerosol properties from METIS and photometer is presented in Fig. 6 for the period of 17 July to 20 July 2022 when the dust and smoke particles were detected up to 6 km altitude. From height-temporal variations in Fig. 6 panels
(a) to (d), two periods can be distinguished during the event. On 17 July 2022, a predominant layer of ~1.5 km width and quite homogeneously distributed is observed between 2 and 5 km asl, in contrast to the three compacted layers detected from 18 July until 19 July 2022 12:00 UT. Contrariwise to the complexity observed with the lidar, the temporal series from the photometer are quite stable (Fig. 6e).

For the first period on 17 July 2022, retrieved properties are on average $0.10 \pm 0.01$ for VLDR, $68 \pm 12$ Mm$^{-1}$ ($76 \pm 34$ sr) for
extinction (LR) at 532 nm and $44 \pm 9$ Mm$^{-1}$ ($33 \pm 14$ sr) for extinction (LR) at 808 nm, respectively, for the layer at 3-4.5 km asl. Only data from 18:00 to 24:00 are considered for 808 nm. During the second period on 18-19 July 2022, the layer from the day before now reduced to 0.5 km width is descending from 3 km towards 1 km asl accompanied by 2 separated layers above it. In particular, we focus our attention on the afternoon of 18 July 2022 to early morning of 19 July 2022, where quite stable $AOD_{ph}$ and $EAE_{ph}$ are observed. LR is on average $47 \pm 6$ sr and $35 \pm 8$ sr at 532 nm and 808 nm, respectively. The second layer (2.4-3.2
km asl) shows lower VLDR values of $0.07 \pm 0.01$ and higher extinction ($50 \pm 3$ Mm$^{-1}$sr$^{-1}$ at 532 nm and $36 \pm 2$ Mm$^{-1}$sr$^{-1}$ at 808 nm) than the other 2 layers. The third layer (3.2-4.5 km asl) is, in comparison, characterized by higher VLDR ($0.12 \pm 0.02$) and lower extinction ($40 \pm 2$ Mm$^{-1}$sr$^{-1}$ at 532 nm and $25 \pm 1$ Mm$^{-1}$sr$^{-1}$ at 808 nm). VLDR values are similar to those observed towards the end of the pure dust event presented in Sect. 4.2.1. Towards 12:00 UT on 19 July 2022, the 3 layers disappear while the boundary layer height increases and probably mixes with the layer closer to the ground.

The VSD distributions during the event (Fig. 7) showed the predominance of three aerosols sizes, one in the fine mode centered at 0.11 μm radius, and two in the coarse mode centered at 1.7 μm and 5 μm. On 18 July 2022 (Fig. 7b) 5 VSD were retrieved, all having higher concentration than the day before (Fig. 7a), only one VSD in the morning is offset to higher values (0.15 μm) for the fine mode peak. On 19 July 2022 (Fig. 7c), 7 VSD were retrieved, 4 of them in the morning showing the same shape as the ones from 18 July. The rest of the VSD show higher contribution at 5 μm size, representing the conditions after 15:00 on 19 July

which correspond to a drop on the AOD values and the vanishing of the layers. Therefore, the presence of both smoke (fine mode) and dust (coarse mode) aerosols can be confirmed during the entire event (Fig. 7), with mainly two different stages in the aerosol vertical distributions (Fig 6).

For comparisons of METIS and LILAS, averaged profiles between 01:00 to 03:00 UT on 19 July 2022 are used (when Raman measurements from LILAS are available). The lunar measurements available are averaged during the same time period to constrain the inversion for METIS. During this event, LILAS lidar got affected by the extreme environmental conditions, so a higher incomplete overlap is acknowledged and we will not consider retrievals comparisons below 1.7 km. Also, METIS overlap corrections induce errors in the first 2 km of the RCS at 532 nm, from 3 % at 2 km going towards 20 % at 600 m. For RCS at 808 nm the influence of overlap error goes from 5 % at 600 m towards 20 % at 100 m. For retrievals using both RCS, values are therefore considered constant below 600 m. Once again, PLDR, EAE and CR values are not shown when the aerosols backscatter at 532 nm is less than 0.3 $Mm^{-1}$ $sr^{-1}$ and at altitudes below 600 m.

Backscatter coefficients (Fig. 8a) and depolarization ratios (Fig. 8c) comparisons show good agreement between both lidars above 2 km asl with an obvious influence of the vertically-constant LR assumption on METIS for the retrieval of backscatter profiles. The extinction coefficients (Fig. 8b) and consequently the EAE (Fig. 8d) are the most impacted (LR values of 38 ± 2 sr for 532 nm and 40 ± 2 sr for 808 nm), showing the limitation of the inversion method under complex scenarios. However, VLDR and PLDR values retrieved from METIS are highly sensitive to the change of dust-smoke composition within the layers. The first layer between 1.6-2 km asl and the third layer between 3.5-5 km asl showed PLDR (VLDR) values in average 0.20 ± 0.02 (0.09 ± 0.01) and 0.27 ± 0.03 (0.12 ± 0.01), respectively, both layers with insights of dust predominant presence. In contrast, the second layer (2.4 - 3.2 km asl) yields the unique presence of smoke aerosols with PLDR (VLDR) of 0.09 ± 0.01 (0.05 ± 0.01), which are in accordance with reported values of fresh smoke transported 1 day from source (Balis, 2003; Ansmann et al., 2009; Tesche et al., 2009b; Alados-Arboledas et al., 2011). Therefore, EAE values (Fig. 8d) are expected to be higher than 1 for the second layer, which is not the case due to the use of vertically-constant LR. Moreover, ACR values directly derived from METIS measurements are influenced by the aerosol attenuation but are still sensitive to the different layers, in contrast to the CR profile derived from the inversion. Furthermore, the limitations discussed can be reduced by adding iterative processes to retrieve layer independent LR, as proposed by (Lu et al., 2011).

Thanks to the operational improvements for the roof window of METIS, a reduced relative VLDR bias of 12 % with respect to LILAS is achieved. The results shown here are evidence of the relevant upgrades in the CE376 system relative to the previous model CE370 for an enhanced aerosols characterization. Furthermore, the algorithmic assessment presented in this first part of the results, provided us with necessary tools to evaluate the data acquired during the FIREX-AQ campaign.

## 5 Mobile exploratory platform

On this work, we presented the dual wavelength CE376 lidar that gives access to valuable information on the particles size with the measurements at 2 wavelengths and on aerosols shape using the depolarization measurements. The capabilities of the instrument regarding continuous monitoring and characterization of aerosols have been presented in Sect. 4. Furthermore, the CE376 lidar is automatic, lightweight and compact, which are favorable attributes for its installation on reduced space. In comparison with bulky high power lidars, the CE376 does not demand constant maintenance or high-power consumption. Therefore, the CE376 has been proposed to continue the developments on remote sensing mobile exploratory platforms.



In this section, we present a first dataset obtained with the CE376 lidar and photometer on-board a mobile platform during the FIREX-AQ campaign in summer 2019. The general description of the campaign's mobile component is presented in Sect. 5.1 with an overview of the spatio-temporal variability of smoke optical properties observed during the campaign (Sect. 5.1.1). Combined mobile-stationary measurements during William Flats Fire are presented in Sect. 5.2 through case studies.

**5.1 FIREX-AQ Dragon Mobile Unit**

The extensive field campaign FIREX-AQ, led by NOAA and NASA, was created with broad science targets (Warneke et al., 2023), mainly focusing on investigating the chemistry and transport of smoke from wildfires and agricultural burning with the aim of improving weather, air quality and climate forecasts. FIREX-AQ has been organized during summer 2019 over the Northwest states of US, where intense wildfires and agricultural fires take place. In order to evaluate and study the smoke properties at the
source and its transport on a local and regional scale, remote sensing instruments were installed in both stationary and mobile DRAGON (Distributed Regional Aerosol Gridded Observations Networks) payloads, in addition to the permanent AERONET sites (Holben et al., 2018). In total, three DRAGON networks were installed in Missoula, Taylor Ranch, and McCall and two mobile units with photometer-lidar were deployed.

The two mobile units called DMU-1 and DMU-2 (Dragon Mobile Unit), both equipped with photometer and lidar, performed on-
road mobile measurements around major fires sources. The installation of the remote sensing instruments in the DMUs followed the design of MAMS (Mobile Aerosol Monitoring System) platform (Popovici et al., 2018). DMU-2 was equipped with CE370 mono-wavelength lidar and PLASMA sun photometer, both tested and used in prior mobile campaigns (Popovici et al., 2018; Hu et al., 2019; Popovici et al., 2022). DMU-1 was equipped with an early version of CE376, two-wavelength polarization lidar, and with the CE318-T sun-sky-lunar photometer (ship-borne CE318-T). Depolarization measurements at 532 nm followed a
configuration with $\varphi=0°$, measuring the parallel component on the PBS transmitted branch, so that $RCS_T(r) = RCS(532 \parallel, r)$, $RCS_R(r) = RCS(532 \perp, r)$, $\delta*(r)=RCS(532 \perp, r)/RCS(532//, r)$ and the VLDR defined by Eq. (4). The measurements were taken through an open hatch in the rooftop of the vehicles, so no influence of a window on the depolarization measurements. The temperature control inside both mobile units was not possible during mobile measurements (only using the car's air conditioning), so stationary and in movement measurements were alternated with pauses to preserve the instruments performance, especially
during daytime when extremely high temperatures and dry conditions were met. Particularly for the 532 nm channels of the CE376 lidar, the overlaps were affected by the daily evolution of temperatures varying some days from 15 °C during nighttime to 40 °C during daytime. Therefore, only quality-assured data are considered for the inversion scheme in this work. Moreover, the temperature effect was accounted on the overlap correction, from where relative errors of 10 % at 2 km going to 30 % at 400 m are estimated and propagated on the derived aerosol properties.

**5.1.1 Overview of smoke optical properties distribution**

Both DMUs performed measurements along the roads around the major fire sources. Although the extreme conditions, such as high temperatures, topography and the presence of thick smoke plumes, limited the performance of the instruments, we were able to investigate smoke optical properties close to the source. A general overview of the column-integrated optical properties during the campaign is provided by photometer mobile observations around 7 fires sources (Table 2). Measurements in and out of smoke
plumes within ~150 km from the fires are taken into account for the average values presented. The high concentration of fine mode aerosols (expected for fresh smoke) is detected at a regional level, with $EAE_{ph}(440/870)$ always higher than 1.3, and varying 5%



from the averages at each fire. On the other hand, measured $AOD_{ph}(440)$ are varying up to 40 % from the averages at each fire, showing a non-homogeneous distribution of aerosols around the source.

Adding measurements from the lidars system, a more elaborated study of the spatio-temporal distribution of aerosols properties can be addressed. Therefore, optical properties retrieved from lidar and photometer measurements are presented in Sect. 5.2 through case studies during William Flats Fire.

**5.2 William Flats Fire at WA, USA** *(6 to 7 August 2019)*

The western US was affected by a persistent deep trough of low pressure in the months prior to FIREX-AQ resulting in elevated soil/vegetation moisture when the fire season began, which controlled the regional fires spread. However, during the first days of the campaign (22 July-5 August 2019), high pressure (anticyclone) weather conditions controlled the moisture transport in the mid-troposphere with wide spread of cloud cover and thunderstorms. Combined with dry conditions in the lower troposphere, precipitation normally evaporated before reaching the ground, allowing the ignition of various fires due to lightning strikes. A low-pressure trough approaching from the West (W) on 6-9 August 2019 broke the high-pressure ridge increasing gradually surface winds speed. William Flats fire, hereafter denominated simply as WFF, in the North-East (NE) of Washington state was in particular controlled by the unique synoptic weather conditions, with fire spread and smoke release progressively increasing as the low-pressure approached. A more detailed description of the synoptic meteorological conditions dominating the campaign can be found in Warneke et al. (2023). Moreover, a camping base has been installed at Fort Spokane (47.905° N, 118.308° W, 430 m a.sl.), which is located on the East (E) side of WFF at ~15 km from the source and separated by the Columbia River.

Mobile observations from selected on-road trajectories completed during 6-7 August 2019 are taken into account to reveal the distribution of aerosols properties around the active WFF. Thus, the GPS track of lidar measurements and the photometer observations from both DMU-1 and DMU-2 are displayed in Fig. 9. The selected trajectories (T) for DMU-1 (T1 to T4), in the top panel, and for DMU-2 (T1 to T5), in the bottom panel, are represented by different symbols. The time used to cover each of them is indicated on the legend and also on top of the maps, all times are in UT (Local time + 7h). In addition, the $AOD_{ph}$ values at 440 nm from both photometers are given by the symbol size, and $EAE_{ph}$ values at 440-870 nm are color-coded. To simplify the reading of this section, $AOD_{ph}$ refer to $AOD_{ph}$ values at 440 nm and $EAE_{ph}$ to $EAE_{ph}$ values at 440-870 nm when wavelengths are not specified. The fire ignition point is indicated on the maps with a red star symbol and Fort Spokane is pointed with a blue arrow. The extension of the active fire for each day are represented with the thermal anomalies, or hot spots, from the satellite-based sensor MODIS (Moderate Resolution Imagin Spectroradiometer). The MODIS Thermal anomalies product is derived from the Terra and Aqua satellites and it is available to the public through NASA Worldview (https://wvs.earthdata.nasa.gov, last access: 23 October 2023).

The CE318-T photometer aboard DMU-1 was adapted and used for ship-borne type of mobile measurements, i.e., for slow motion, before the campaign. Therefore, some difficulties were faced when using a car, especially due to the velocity and the complexity of the terrain and roads. The sun-tracking and geo-location communication were not fast enough for these particular conditions. As a solution, stationary measurements of 5 to 15 minutes were performed along the DMU-1 trajectories to increase the density of observations with CE318-T photometer. On the other hand, PLASMA sun-photometer was able to successfully perform on-road observations, with difficulties mainly due to the presence of mountains when sun elevations are low and in presence of dense smoke plumes. Differences on both photometer performances are clear in Fig. 9. In general, both DMU-1 and DMU-2 observations during 6-7 August 2019, show the predominance of fine aerosols with $EAE_{ph}$ values always higher than 1.4, as well as high



variability of aerosols distribution with AOD$_{ph}$ ranging from 0.1 to 1.1. For further interpretation of the photometer mobile
observations, it is convenient to mention the solar azimuth during the WFF. Hence, at sunrise (~13:40 UT) the azimuth is 68°
(NEE), at solar noon (~21:00 UT) is 180.4° (S) with elevation of 58.7° and at sunset (~04:40$^{+1day}$ UT) the azimuth is 292° (WNW).
In the following sub-sections, the analysis of mobile observations from DMU-1 and DMU-2 for each day are presented.

### 5.2.1 Three-dimensional spatio-temporal variation of smoke properties

On 6 August 2019, WFF was spread to the NE from its ignition point, with hot spots land elevations ranging around 0.7-1.2 km
asl (Fig. 9 panels a and c). Plumes of emitted smoke were mostly moving to E direction with respect to the source. Approaching
to the sunset (~04:40$^{+1day}$ UT), smoke release progressively increased with the temperature rising. Hence, the spatio-temporal
distribution of aerosols along the trajectories for both DMU-1 (top panel) and DMU-2 (bottom panel) are presented in Fig. 10. For
each trajectory, the 3D spatio-temporal distribution of $\beta_{att}$ at 532 nm is plotted on top of the 3D Digital Elevation Model (DEM)
map of the region. The DEM used is the product 1 arc-second global coverage (~30 m resolution) from Shuttle Radar Topography
Mission (SRTM), available through Earth Explorer interface of United States Geological Survey (https://earthexplorer.usgs.gov/,
last access: 23 October 2023). Moreover, both $\beta_{att}$ and DEM maps are color coded, each one with its own color bar scale. In the
same way as in Fig. 9 panels (a) and (c), red points represent the thermal anomalies showing the extension of the active WFF
detected on 6 August 2019.

During 6 August 2019, residual smoke in all the trajectories was detected up to 4 km asl and higher AOD$_{ph}$ and EAE$_{ph}$ values were
identified under the presence of dense smoke plumes. The Columbia River acted like an air canal with the prevailing valley winds
in the morning (De Wekker and Kossmann, 2015; Whiteman, 2000), directing a diffused smoke plume northward. The trajectory
DMU-1 T1 (Fig. 10a) covered ~80 km between 17:00 to 20:31 UT along the Columbia riverside going from Fort Spokane to Kettle
Falls (48.60° N, 118.06° W). AOD$_{ph}$ ranged within 0.2-0.3 and EAE$_{ph}$ was higher than 1.6 (Fig. 9a). DMU-2 T1 (Fig. 10c) covered
40 km of the same route between 18:00 to 19:28 UT, starting with 30 min of stationary measurements at Fort Spokane. AOD$_{ph}$
values within 0.3-0.7 and EAE$_{ph}$ above 1.7 were observed (Fig. 9c). During both trajectories, azimuthal solar angles vary from
101° to 153° (E to S), meaning that both photometers were taking measurements towards the E side of WFF against the movement
of the vehicles and limited by the mountain slopes. Hence, both DMUs followed and measured the diffuse smoke plume with one
hour time difference. DMU-2 T1 lidar-photometer measurements indicate an increase on smoke release and accumulation
northward, with higher AOD$_{ph}$ and β$_{att}$ (below 2 km asl) values.

The trajectory DMU-1 T2 (Fig. 10b, also Fig. 9a) was completed from 21:50 to 02:59 UT, i.e., in the afternoon, and covered ~100
km on the way back to Fort Spokane from Kettle Falls, passing through Colville River basin. Hence, the residual smoke well mixed
up to 4 km asl is contained along the valley showing AOD$_{ph}$ varying between 0.3-0.5 and EAE$_{ph}$ of 1.6 (solar azimuth 206° to 292°,
i.e., photometer pointing to E side of WFF towards WFF). Approaching Fort Spokane, the development of a convective smoke
plume was observed (Fig. 10b). One exceptional sampling of the dense smoke plume was possible, at ~01:00 UT and 20 km E
away from the fire, with an AOD$_{ph}$ of 1.1 and EAE$_{ph}$ of 2.2 (Fig. 9a). DMU-2 T2 (Fig. 10d, also Fig. 9c) performed measurements
in the afternoon from 23:00 to 23:48 UT going downwind WFF and covering ~50 km horizontally to E (solar azimuth 228° to
245°, i.e., towards WFF). This trajectory in particular shows how smoke accumulated and settled across the valleys. High AOD$_{ph}$
values above 0.7 and EAE$_{ph}$ above 2 (Fig. 9c) were observed. DMU-2 T3 (Fig. 10.e) also completed during the afternoon (23:50
- 01:05 UT), is covering the return route to Fort Spokane. While it got closer to the source, higher values of $\beta_{att}$ (> 6 Mm$^{-1}$sr$^{-1}$)
were detected from 4 km asl towards ground level. Although no photometer data is available due to presence of the thick smoke



plume, lidar provides a glimpse of the convective smoke plume transect. The smoke plume raised up to 4.2 km asl at 50 km away (horizontally to E) from its source, ~3 km higher than the active fire and above the mountain ridges.

During 7 August 2019, the WFF extended towards E getting closer to the Columbia River ridge, and more hot spots were detected than the day before (Fig. 9b and Fig. 9d). Through the day, smoke convective plumes moved, mostly influenced by the strong

winds, towards E direction and slightly to SE. In the afternoon, black and white ash depositions were reported, in addition to clouds formation observed close to sunset (~04:40$^{+1day}$ UT). At that point, the presence of heavy smoke plumes saturated the lidar signals and restricted photometers measurements close to the source. Therefore, trajectories were performed mostly outside the smoke plumes. Same as for lidar observations presented in Fig. 10, 3D spatio-temporal distributions of $\beta_{att}$ at 532 nm for all the trajectories during 7 August 2019 are presented in Fig. 11.

The trajectory DMU-1 T3 (Fig. 11a) covered ~40 km from Fort Spokane to the S of WFF between 18:00 to 19:58 UT. DMU-1 T4 (Fig. 11b) covered ~70 km of route from S to E side of WFF, between 21:00 to 23:59 UT. For both trajectories, few data points from photometer were collected and might not represent the same conditions for the zenithal lidar measurements. Photometer is looking towards SE to SW from the WFF, against the winds flow. AOD$_{ph}$ ranging between 0.1-0.2 and EAE$_{ph}$ above 1.6 were observed (Fig. 9b) which are indication of low loading of residual smoke on the S region of WFF. Both trajectories seen by the

lidar show no direct influence of the smoke release on the S-SE of WFF and present considerably lower values of β$_{att}$. Nevertheless, similarly to observations on 6 August 2019, a convective smoke plume reaching up to 4 km asl is observed in the afternoon (Fig. 11b).

On the other hand, the trajectory DMU-2 -T4 (Fig. 11c) is covering the NNE of WFF along the Columbia riverside and following the smoke plume. DMU-2 T4 covered ~80 km from Fort Spokane to Kettle Falls, from 16:49 to 18:39 UT and with AOD$_{ph}$ ranging

within 0.1-0.3 and EAE$_{ph}$ 1.6-1.8, higher AOD values being measured closer to the fire. This time, the vertical extent of the smoke plume is ~200 m higher and it is denser than the day before. But in the same way as the day before, the Columbia River is the main driver of the channeling effect of the smoke towards the N in the morning. The trajectory DMU-2 T5 (Fig. 11d and also Fig. 9d) covered ~200 km between 18:40 to 23:40 UT from Kettle Falls (80 km NNE from WFF) towards Davenport (47.65° N, 118.15° W, ~40 km SE of WFF) going through valleys and returning to Fort Spokane. Along the way, DMU-2 measured residual smoke

accumulated in the NE valley basins, with AOD$_{ph}$ around 0.3 and EAE$_{ph}$ of 1.6-1.8. In addition, residual smoke, SE of WFF, was measured with lower values of AOD$_{ph}$ around 0.1-0.3 and EAE$_{ph}$ 1.5-1.6. During this transect the DMU-2 crossed 2 times the smoke plume, one at 21:20-21:23 UT 40 km downwind WFF, and the second time at 23:00 UT 15 km away from the WFF. From the DMU-2 T5 3D aerosol distribution (Fig. 11d) and photometer (Fig. 9d), one can see the effect of the diffuse smoke from WFF on the NE region, characterized by its mountains and valleys.

The complex topography combining with the prevailing synoptic conditions (low pressure trough approaching from the W) have important effects on the development of fire (Whiteman, 2000). While in the morning the river basin acted almost independently, channeling smoke northward, we noticed how the evolving boundary layer is coupled to the mountain winds systems. The diffused smoke is mixed and subsided along the valleys, with higher aerosols loading closer to the fire downwind. Moreover, fire emissions get stronger while temperatures rise up, permitting the convective loft of the smoke above the mountain ridges. On 7 August 2019,

the convective smoke evolved into the formation of pyrocumulus clouds. For further analysis, in the following section we present aerosol retrievals of selected datasets from the trajectories presented here.





### 5.2.2 Aerosols properties for selected profiles

From the DMU-1 and DMU-2 trajectories on 6-7 August 2019, selected coincident lidar and photometer data are averaged over 5 to 15 minutes and are used to enhance the aerosols characterization presented so far. The selected times are displayed in Fig. 10
and Fig. 11 by orange arrows in the 3D $\beta_{att}$ quicklook. In Fig. 12, we present the profiles of aerosol properties for each selected dataset differentiated by color. Hence, we show profiles of backscatter, extinction at 532 nm and 808 nm, and profiles of PLDR, EAE and ACR. For the lidar retrievals, data below 400 m is considered constant due to high uncertainties (>30%) on RCS at 532 nm. Molecular coefficients are calculated using radiosonde measurements at Spokane station (47.68° N, 117.63° W) from Wyoming University database (https://weather.uwyo.edu/upperair/sounding.html, last access: 23 October 2023). The detection
limit is defined at SNR=1 for all channels to extract more information, in particular from 808 nm.

Detection limits for 808 nm and 532 nm cross polarized channels from CE376 are below 2 km and 3-4 km, respectively, due to high solar background. Nevertheless, we were able to study the diffuse smoke plume transported along the Columbia River with retrievals profiles from selected data. The datasets A, attained during DMU-1 T1, is showed in Fig. 12 panels (a) to (g) and B, from DMU-2 T1, is showed in Fig. 12 panels (a) and (c). The dataset A corresponds to the averaged CE376 lidar data from 18:10
to 18:25 UT on 6 August 2019, located 40 km away to the NNE of WFF. AOD$_{ph}$ from CE318-T photometer were 0.28 and 0.13 at 532 nm and 808 nm, respectively, EAE$_{ph}$(532/808) was 1.76 and retrieved AOD$_{est}$ at 808 nm is 0.1. The smoke plume is identified at 1-1.3 km asl with maximum values of extinction at 1.14 km asl. Thus, extinction values of $370 \pm 70$ Mm$^{-1}$ (with LR=35 ± 1 sr) at 532 nm (Fig. 12c), and $207 \pm 20$ Mm$^{-1}$ (with LR= 57 ± 4 sr) at 808 nm (Fig. 12d) were observed. Other aerosol properties inside the smoke plume were $0.06 \pm 0.04$ for PLDR (Fig. 12e), $1.2 \pm 0.5$ for EAE (Fig. 12f) and $0.5 \pm 0.3$ for ACR (Fig. 12g). On the
other hand, dataset B corresponds to averaged CE370 lidar data from 19:05 to 19:15 UT on 6 August 2019, ~1 h after the dataset A was obtained. Dataset B is located 25 km to the NNE away from WFF, with values of 0.35 for AOD$_{ph}$ at 532 nm and 1.7 for EAE$_{ph}$ (440/870). The smoke plume is identified at 1.6-1.9 km asl with maximum values of extinction at 1.71 km asl. Values of $380 \pm 20$ Mm$^{-1}$ (with LR=39 ± 1 sr) for extinction at 532 nm were retrieved. The identified smoke plumes for both datasets are almost the same, except for the altitude. The higher extinction below 1 km asl for dataset B is related to the increase of smoke
released through the day. Moreover, a layer of residual smoke at 2-3 km asl is detected for both cases, with less intensity for dataset B but still noticeable. PLDR in the residual layer ($0.08 \pm 0.02$) is in agreement with reported values of fresh smoke transported one day from source (Balis, 2003; Ansmann et al., 2009; Tesche et al., 2009b; Alados-Arboledas et al., 2011). Despite the high uncertainties that are attached to the profiles in the first hundreds of meters, ACR values (Fig. 12g) suggest the presence of bigger aerosols in the smoke plume at 1 km asl than in the residual layer at 2-3 km asl, in the same way as EAE. The observed bigger
aerosols could be related to the release of fine-ash particles (sizes of 1 μm-2 μm) within the smoke plume (Adachi et al., 2022).

The dataset C showed in Fig. 12 panels (h) to (n), obtained during DMU-1 T2, corresponds to averaged CE376 lidar data from 00:40 to 00:50 UT, toward sunset on 6 August 2019. This dataset, located 20 km E of WFF, is particularly interesting because it provides information on the convective smoke plume. Values of 1.54 and 0.61 for AOD$_{ph}$ at 532 nm and 808 nm, respectively, were detected by the photometer, as well an EAE$_{ph}$(532/808) of 2.25, and retrieved AOD$_{est}$ at 808 nm of 0.18 (below the smoke
plume). The convective plume is identified at 3-4.3 km asl, with maximum values of extinction at 3.57 km asl (Fig. 12j). Thus, $1270 \pm 330$ Mm$^{-1}$ (with LR= 82 ± 2 sr) for extinction at 532 nm was observed. Inside the plume, a decrease of the PLDR (Fig. 12l) from $0.05 \pm 0.01$ to $0.03 \pm 0.01$ is detected, in addition to values progressively increasing from $0.4 \pm 0.1$ to $0.9 \pm 0.1$ for ACR (Fig. 12n). Both parameters suggest the predominance of big spherical particles towards the smoke layer top, which could be related to the fast increase in the coating mass of soot particles within minutes from emission. In contrast, dataset D showed in Fig. 12 panels





(o) to (u), located 25 km S of WFF (21:00 to 21:09 UT on 7 August 2019), and dataset E showed in Fig. 12 panels (o) and (q), located 60 km NE of WFF (20:00 to 20:30 UT on 7 August 2019), present residual smoke. Both datasets have values of 0.13 for $AOD_{ph}$ at 532 nm. The dataset D shows a residual layer extending up to 4 km asl, with average values of $44 \pm 17$ Mm$^{-1}$ (with LR= $37 \pm 3$ sr) for extinction at 532 nm (Fig. 12q), and $28 \pm 15$ Mm$^{-1}$ (with LR=$87 \pm 15$ sr) at 808 nm (Fig. 12r). Moreover, PLDR is $0.09 \pm 0.03$ (Fig. 12s), EAE is $1.5 \pm 0.3$ (Fig. 12t) and ACR is $0.3 \pm 0.1$ (Fig. 12u). One has noticed that ACR values are constant

within the residual layer, suggesting that smoke is well mixed. Dataset E shows the residual smoke in the NE side of the WFF is going up to 3 km asl with a LR of $73 \pm 7$ sr, higher than for dataset D.

## 6 Summary and Conclusions

In this study, we presented the enhanced capabilities of the CIMEL CE376 lidar, a compact dual-wavelength depolarization elastic lidar, for the assessment of spatiotemporal variability of aerosol properties, especially when deployed aboard moving platforms

and co-located with a photometer. Our approach involved a modified two-wavelength Klett inversion constrained by photometer measurements, optimizing the use of synergetic observations. Comprehensive algorithmic and instrumental assessments, including improvements in continuous depolarization measurements, were conducted at the ATOLL observatory. Our findings were organized into two primary parts: with the aerosol properties retrieved from the case studies at the ATOLL observatory in Lille, France (Sect. 4) and around the William Flats Fire in Northwestern US during the FIREX-AQ campaign (Sect. 5). Aerosol optical

properties obtained in both sections are summarized in Table 3.

Both algorithmic and instrumental assessments of CE376 were tested through case studies (Sect. 4), encompassing events involving aged dust, as well as mixed dust and smoke over Lille (Table 3). Despite operational limitations, we achieved a relative VLDR bias of 12% compared to LILAS Raman lidar and we showcased CE376's ability for continuous monitoring of aerosol properties. The limitations of our retrieval approach were also evaluated, owing mainly to the assumption of a constant LR in the atmospheric

column, where EAE and CR are the most affected. The unusual event of stratified dust and smoke transported over Lille highlights the importance of depolarization measurements for aerosol typing within the different aerosol layers, demonstrating CE376's reliability even in challenging scenarios.

We also presented for the first time ground-based lidar and photometer mobile observations, mapping smoke aerosols properties near the source during the FIREX-AQ campaign in 2019 (Sect. 5). Our study focuses on William Flats Fire (WFF) in Washington

state, which presented unique and challenging environmental conditions for the exploratory platforms. The 3D mapping of lidar and photometer observations enabled the identification of aerosol properties in diffuse, convective, and residual smoke layers near the WFF (Table 3). The study revealed the capabilities of CE376 aboard mobile platforms to characterize the smoke aerosols optical properties. At the same time, we acknowledged the limitations of the CE376 lidar and photometer in harsh environmental conditions (complex topography, high temperatures, thick smoke plumes).

In perspective, with the demonstrated versatility of the CE376 lidar for monitoring aerosol properties, we look ahead for bridging observational gaps within networks. Therefore, upcoming mobile campaigns (aboard ship cruises, trains, and cars) and permanent sites in the southern hemisphere are planned to include the upgraded, more robust version of the CE376 lidar. The installation of a CE376 lidar aboard Marion Dufresne research vessel, in the framework of MAP-IO, is planned in 2024. Moreover, the Polar POD (https://www.polarpod.fr/, last access: 24 October 2023), a floating scientific platform that will circle the Earth around Antarctica,

will include a CE376 automatic lidar, along with several scientific instruments to be installed. Additionally, ongoing research involving advanced retrieval methods like GRASP (Generalized Retrieval of Aerosol and Surface Properties), combining spectral



AOD and downward sky radiance from CE318-T photometers and RCS at two wavelengths from CE376 are under way. These advancements mark significant steps in enhancing our understanding of aerosol dynamics and environmental monitoring.

**Data availability**

Data from photometer are available at AERONET website (https://aeronet.gsfc.nasa.gov, last access: 23 October 2023). Radiosonde data are accessible at the Wyoming University database (https://weather.uwyo.edu/upperair/sounding.html, last access: 23 October 2023), and Meteo-France database (https://donneespubliques.meteofrance.fr, last access: 23 October 2023). The data of DEM from SRTM are available at Earth Explorer interface of USGS (https://earthexplorer.usgs.gov/, last access: 23 October 2023). The MODIS thermal anomalies product is available at NASA Worldview (https://wvs.earthdata.nasa.gov, last

access: 23 October 2023). Lidar data used in this paper are available upon request to the corresponding author.

**Authors contributions**

MFSB analyzed the CE370 and CE376 lidar data, prepared the figures and wrote the manuscript. PG, SV and IEP supervised the work and contributed to the writing of the manuscript. PG, SV, IEP, LB, BH and BT designed and conceptualized the project of lidar and photometer mobile applications. PG, IEP, LB, EB, QH and TP conceived and performed the experiments at ATOLL

platform. MFSB developed the CE376 algorithmic assessments initiated by IEP. IEP, TP, LP, MFSB and EB supported instrumental assessments of CE376 lidar. QH, TP performed the experiments with LILAS and QH analyzed the data. FD and QH developed and supported LILAS algorithms. PG, IEP, LB, TP, GD, LP, BH, AL, ALR and DG conducted the experiments and supported the installation of instruments aboard DMU-1 and DMU-2 during FIREX-AQ.

**Competing interests**

The authors declare no conflict of interest.

**Acknowledgements**

The authors would like to thank ANRT (Association Nationale Recherche Technologie) France and CIMEL Electronique for supporting the research development in the framework of CIFRE (n° 2020/0442) thesis of MFSB. We acknowledge the CaPPA project (Chemical and Physical Properties of the Atmosphere), funded by the ANR (French National Research Agency) through

the PIA (Programme d'Investissement d'Avenir) under contract "ANR-11LABX-0005-01", the Regional Council "Hauts-de-France" (project CLIMIBIO) and the FEDER (European Funds for Regional Economic Development) for the financial support. We also thank H2020/GRASP-ACE Marie-Curie project for funding. The authors acknowledge generous support from the NOAA AC4 program from FIREX Firelab and FIREX-AQ. The authors thank the AERONET (NASA GSFC and the Service National d'Observation PHOTONS/AERONET-EARLINET) and Cimel Electronique as well as NASA AERONET staff for coordination

with the US Forest Service, Fire Chemistry Lab and the University of Idaho Taylor Ranch Wilderness School staff for set up maintenance and decommission of the stationary DRAGON networks.

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



**Table 1.** System specifications for the mobile lidars. * CIMEL CE370 is no longer commercially available. ** Systems used in this work had higher pulse energy.

|  | CIMEL CE370* | CIMEL CE376 GPN |  |
|---|---|---|---|
| Wavelength | 532 nm | 532 nm | 808 nm |
| Laser source | Frequency doubled Nd:YAG | Frequency doubled Nd:YAG | Pulsed laser diode |
| Pulse energy | 20 uJ | 5-10 uJ (15-20 µJ) ** | 3-5 uJ |
| Repetition rate | 4.7 kHz | 4.7 kHz | 4.7 kHz |
| Emission/Reception (E/R) | Coaxial | Biaxial | Biaxial |
| Telescope (E/R) | Galilean | Galilean | Galilean |
| Diameter (E/R) | 200 mm | 100 mm / 100 mm | 100 mm / 100 mm |
| Half Field of View (E/R) | 55 µrad | 100 µrad / 120 µrad | 240 µrad / 330 µrad |
| Depolarization | No | Yes | No |

**Table 2.** Overview of photometer measurements embarked on-board DMU-1 (CE318-T) and DMU-2 (PLASMA). Averaged measurements
around 7 fires sources during the FIREX-AQ campaign.

| Fire Name | Location (State) | Dates | AOD$_{ph}$ (440) | EAE$_{ph}$ (440-870) |
|---|---|---|---|---|
| Pipeline | 46.83° N, 120.52° W (WA) | 25-28 July, 2019 | 0.17±0.06 | 1.55±0.08 |
| Shady | 44.52° N, 115.02° W (ID) | 29-31 July, 2019 | 0.21±0.01 | 1.90±0.04 |
| Beeskove | 46.96° N, 113.87° W (MT) | 31 July, 2019 | 0.25±0.01 | 1.84±0.03 |
| William Flats | 47.94° N, 118.62° W (WA) | 05-09 August, 2019 | 0.45±0.34 | 1.83±0.13 |
| Nethker | 45.25° N, 115.93° W (ID) | 13-20 August, 2019 | 0.20±0.10 | 1.32±0.10 |
| Granite Gulch | 45.18° N, 117.43° W (OR) | 20-22 August, 2019 | 0.26±0.11 | 1.44±0.08 |
| 204 Cow | 44.29° N, 118.46° W (OR) | 23-29 August, 2019 | 0.70±0.48 | 1.84±0.21 |





**Table 3** Overview of the aerosol properties retrieved from CE376 lidar and CE318-T photometer for the case studies presented in this work. The
estimated uncertainties are in parenthesis. For observations at ATOLL platform, aerosols properties are specified for each layer detected at both
case studies, aged dust (L1, L2) and dust smoke (L1, L2 and L3). For FIREX-AQ campaign, the position with respect to WFF is included.
*Aerosol properties retrieved from CE370 lidar and PLASMA photometer.

| Site | | ATOLL, France | | | FIREX-AQ William Flats fire (USA) | | |
|---|---|---|---|---|---|---|---|
| **Aerosol type** | | **Aged dust** | **Mixture dust+smoke** | **Smoke** | **Diffuse smoke** | **Convective smoke** | **Residual Smoke** |
| **Altitude asl [km]** | | L1: 1.5-3 L2: 3.3-4.7 | L1: 1.6-2 L3: 3.5-5 | L2: 2.4-3.2 | 1-1.3 (40 km NNE) | 3-4.3 (20 km E) | 1.2-4 (25 km S) * 0.9-3 (60 km NE) |
| **LR [sr]** | **532** | [L1, L2] 54 (3) | [L1, L3] 38 (2) | [L2] 38 (2) | 35 (1) | 82 (2) | 37 (3) * 73 (7) |
| | **808** | [L1, L2] 69 (4) | [L1, L3] 40 (2) | [L2] 40 (2) | 57 (4) | - | 87 (15) |
| $\alpha_a$ **[Mm$^{-1}$]** | **532** | [L1] 61 (14) [L2] 43 (3) | [L1] 47 (3) [L3] 34 (2) | [L2] 54 (3) | 370 (73) | 1270 (330) | 45 (17) * 54 (9) |
| | **808** | [L1] 52 (9) [L2] 35 (6) | [L1] 36 (2) [L3] 28 (1) | [L2] 43 (2) | 207 (20) | - | 28 (15) |
| $\delta^v$ | **532** | [L1] 0.15 (0.02) [L2] 0.12 (0.02) | [L1] 0.09 (0.01) [L3] 0.12 (0.01) | [L2] 0.05 (0.01) | 0.04 (0.02) | 0.03 (0.01) | 0.05 (0.01) |
| $\delta^p$ | **532** | [L1] 0.36 (0.05) [L2] 0.36 (0.05) | [L1] 0.2 (0.02) [L3] 0.27 (0.03) | [L2] 0.09 (0.01) | 0.06 (0.04) | 0.04 (0.01) | 0.09 (0.03) |
| **EAE (532/808)** | **LID** | [L1] 0.37 (0.09) [L2] 0.5 (0.08) | [L1] 0.65 (0.04) [L3] 0.52 (0.03) | [L2] 0.55 (0.03) | 1.2 (0.5) | - | 1.5 (0.3) |
| | **PH** | 0.23-0.75 | 0.92 | 0.92 | 1.76 | 2.25 | 1.3 * 1.7 |
| **ACR (808/532)** | | [L1] 0.42 (0.05) [L2] 0.38 (0.04) | [L1] 0.49 (0.03) [L3] 0.5 (0.03) | [L2] 0.56 (0.03) | 0.5 (0.3) | 0.6 (0.1) | 0.3 (0.1) |
| **CR (808/532)** | | [L1] 0.69 (0.14) [L2] 0.65 (0.12) | [L1] 0.72 (0.04) [L3] 0.76 (0.03) | [L2] 0.73 (0.03) | 0.4 (0.3) | - | 0.2 (0.1) |
| **Eff. Radius VSD [µm]** | | 1.7 | 1.7 and 5 | 0.1 | - | - | - |





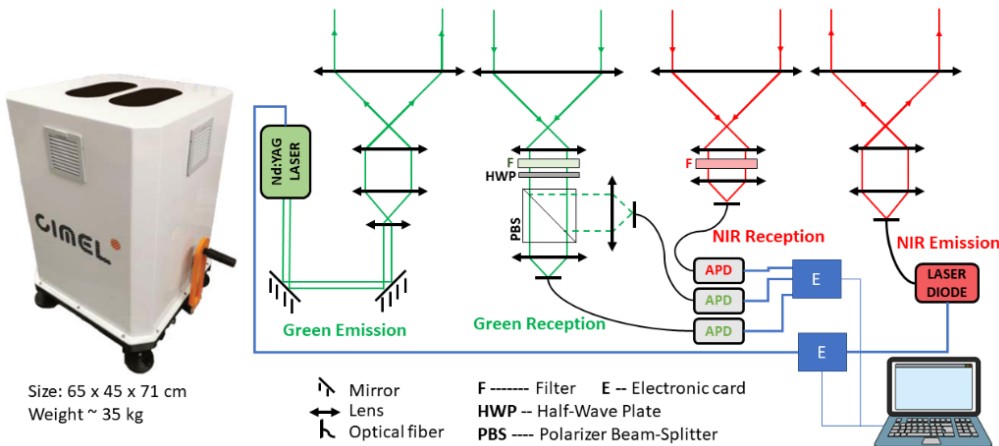


**Figure 1**. CE376 GPN lidar and its 2D design. The optical design of the biaxial systems at 532 nm (Green Emission/Reception) and 808 nm (NIR Emission/Reception), and layout of the control/acquisition system through electronic cards are shown in a simplified plan. Source: https://www.cimel.fr/solutions/ce376/.

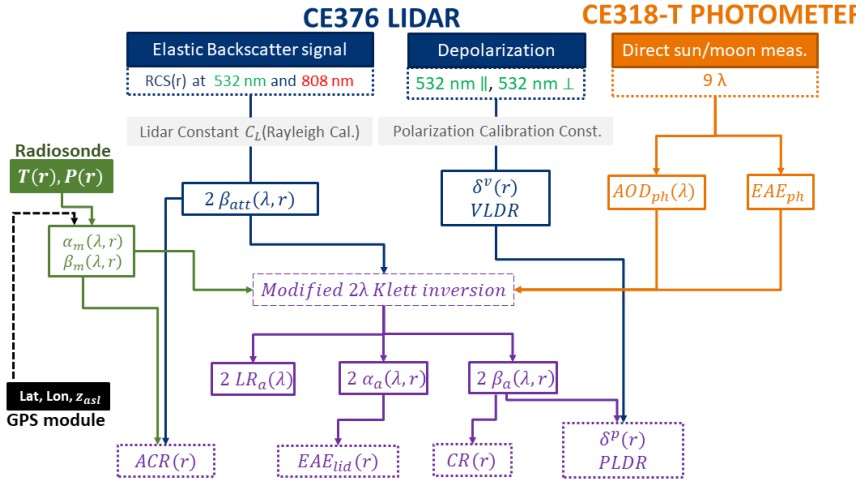

**Figure 2.** Block diagram of the methodology combining measurements from CE376 lidar and CE318-T photometer.





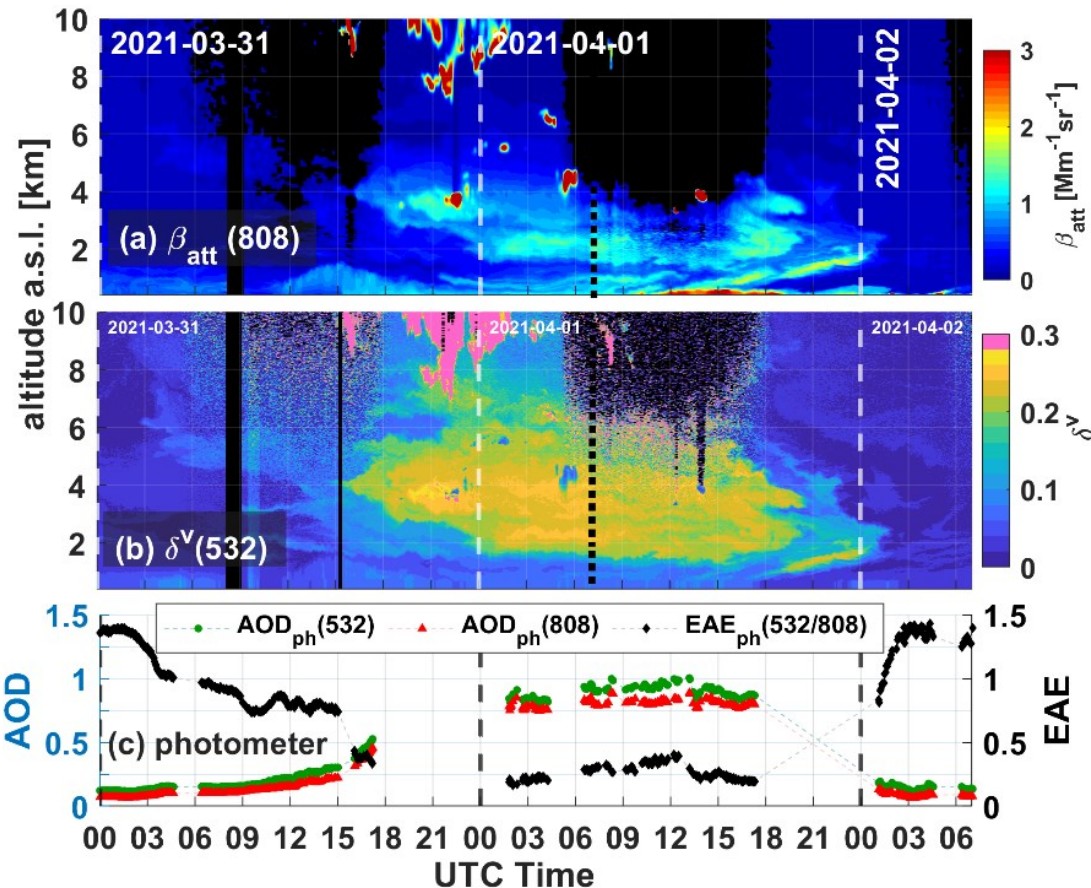

**Figure 3.** Overview of synergetic measurements of METIS lidar and CE318-T photometer during an event of Saharan dust transport from 2021-03-31 to 2021-04-02. Height-temporal variation of (a) $\beta_{att}$ at 808 nm, (b) VLDR at 532 nm, and (c) time series of $AOD_{ph}$ at 532 nm and 808 nm with $EAE_{ph}$(532/808) derived from photometer. Black dashed line in (a) and (b) indicates the change of measurements conditions for METIS lidar.


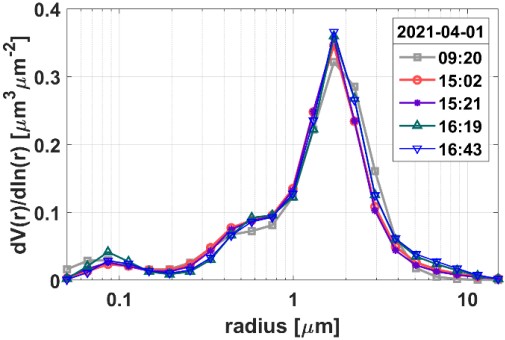

**Figure 4.** VSD derived from CE318-T photometer sky almucantar measurements during 2021-04-01 at ATOLL. Data is level 2 from AERONET version 3 algorithms.



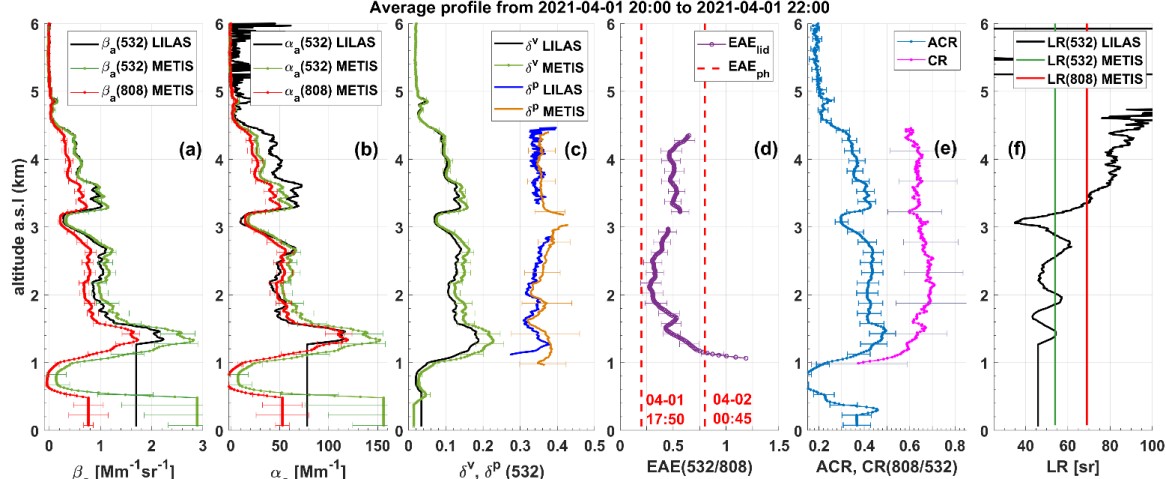

**Figure 5.** Aerosol optical properties retrieved from METIS CE376 lidar and intercomparison with LILAS Raman lidar retrievals for the averaged measurements between 20:00 to 22:00 UT on 2021-04-01. Vertical profiles of (a) Backscatter, (b) Extinction and (f) LR at 532 and 808 nm for METIS and at 532 nm for LILAS, (c) VLDR and PLDR at 532 nm for METIS and LILAS, (d) EAE (532/808) from METIS and the 2 closest values from photometer in red dashed lines, and (e) ACR, CR (808/532) for METIS.





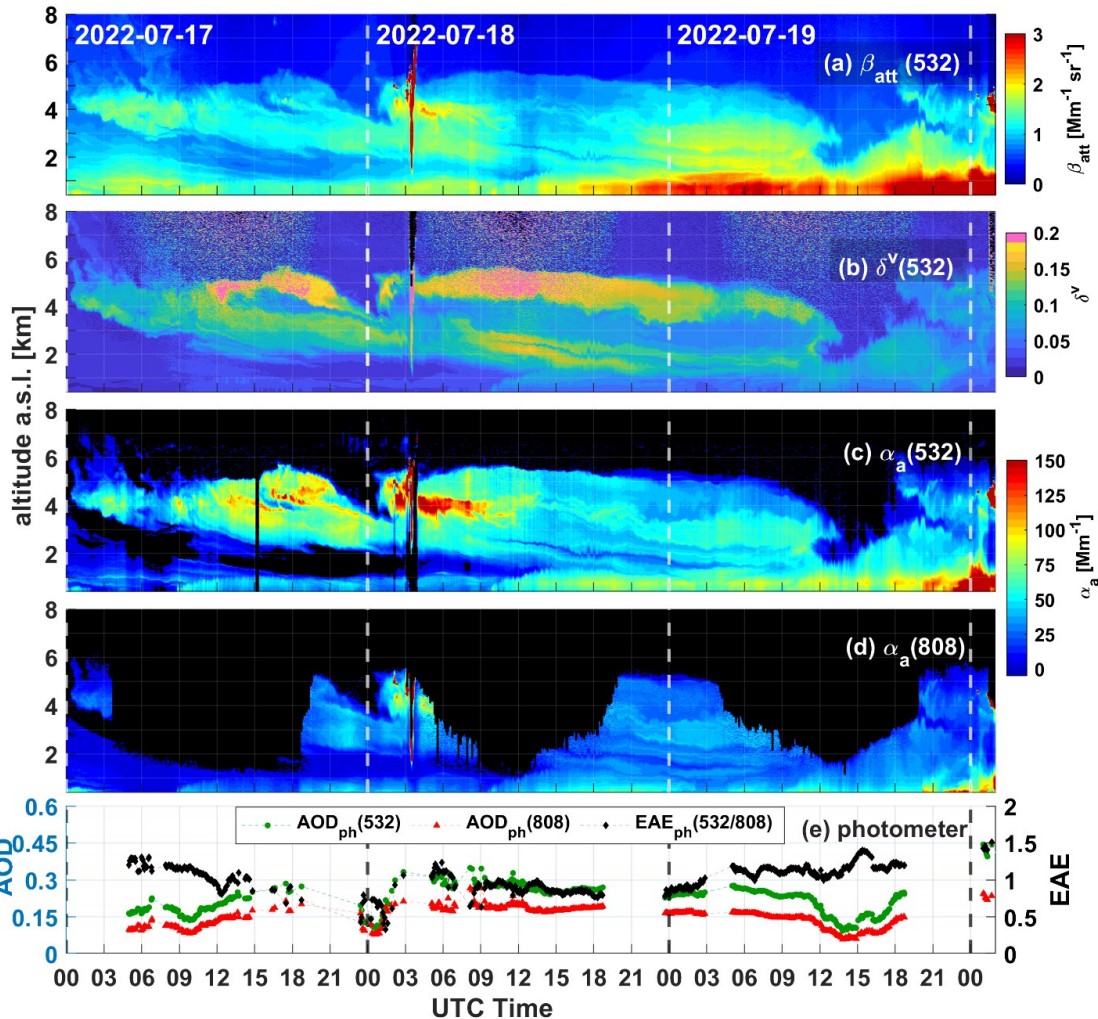

**Figure 6.** Overview of atmospheric optical properties from synergetic measurements of METIS lidar and CE318-T sun/lunar photometer at ATOLL platform from 2022-07-17 to 2022-07-20. Height-temporal variation of (a) $\beta_{att}$ and (b) VLDR at 532 nm, aerosols extinction at (c) 532 nm and (d) 808 nm, and (e) time series of $AOD_{ph}$ at 532 nm and 808 nm with $EAE_{ph}$ 532/808 derived from the photometer.





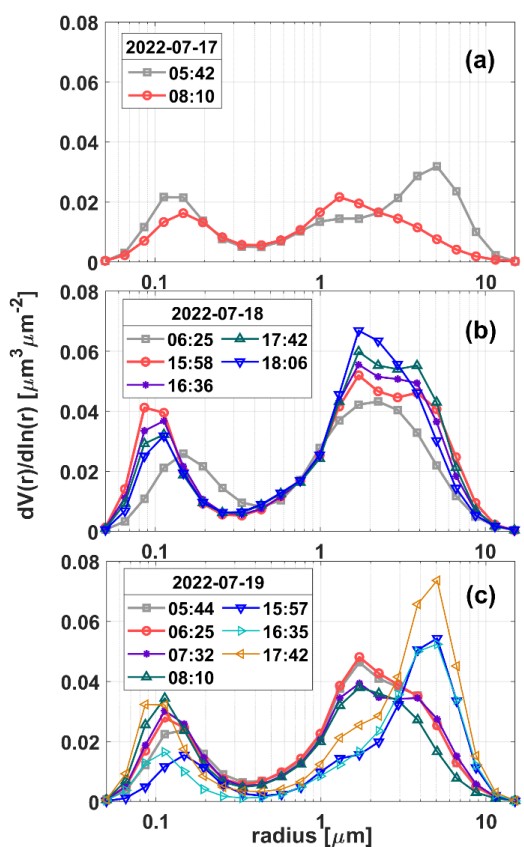

**Figure 7.** VSD derived from CE318-T photometer sky almucantar measurements during (a) 2022-07-17, (b) 2022-07-18 and (c) 2022-07-19 at

ATOLL. Data is level 2 from AERONET version 3 algorithms.

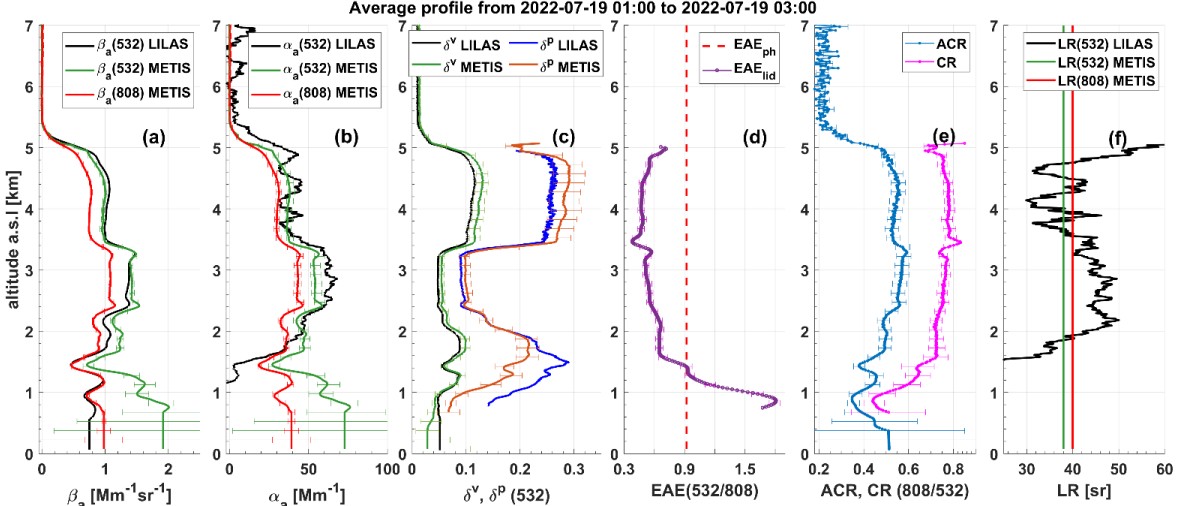

**Figure 8.** Aerosols optical properties retrieved from METIS and comparison with LILAS retrievals, same as Figure 5, but for the averaged

measurements between 01:00 to 03:00 UT on 2022-07-19.



**Figure 9.** Mobile observations around WFF during 2019-08-06 and 2019-08-07 in UT. GPS tracks of DMU-1 and DMU-2 are presented in the top and bottom panel respectively. For each trajectory (T) a different symbol is used. Photometers measurements are presented with color coded symbols, EAE(440/870) represented by the color and AOD(440) by the symbol size. The ignition point of WFF is represented by a red star. The extension of the fire is represented by thermal anomalies from MODIS AQUA/TERRA detected during each day.



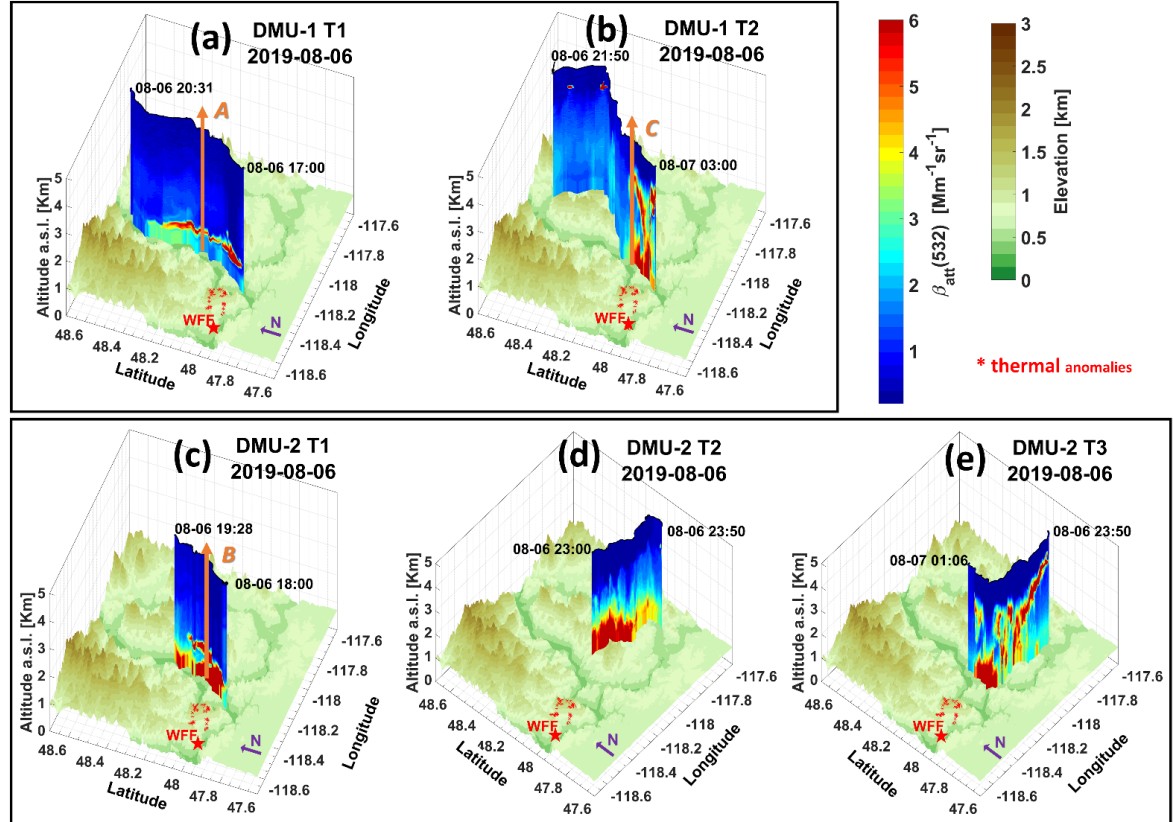

**Figure 10.** Spatial-temporal distribution of total attenuated backscatter at 532 nm for the trajectories during 2019-08-06 from Fig. 9. Trajectories of DMU-1 (CE376 lidar) are presented in the top panel and DMU-2 (CE370 lidar) in the bottom panel. The lidar trajectories are plotted on top DEM from SRTM at 1 Arc-Second resolution (~30 m). The ignition point of WFF is represented by a red star and the extension of the active fire by MODIS thermal anomalies. Orange arrows represent the selected profiles for further analysis in Fig. 12.



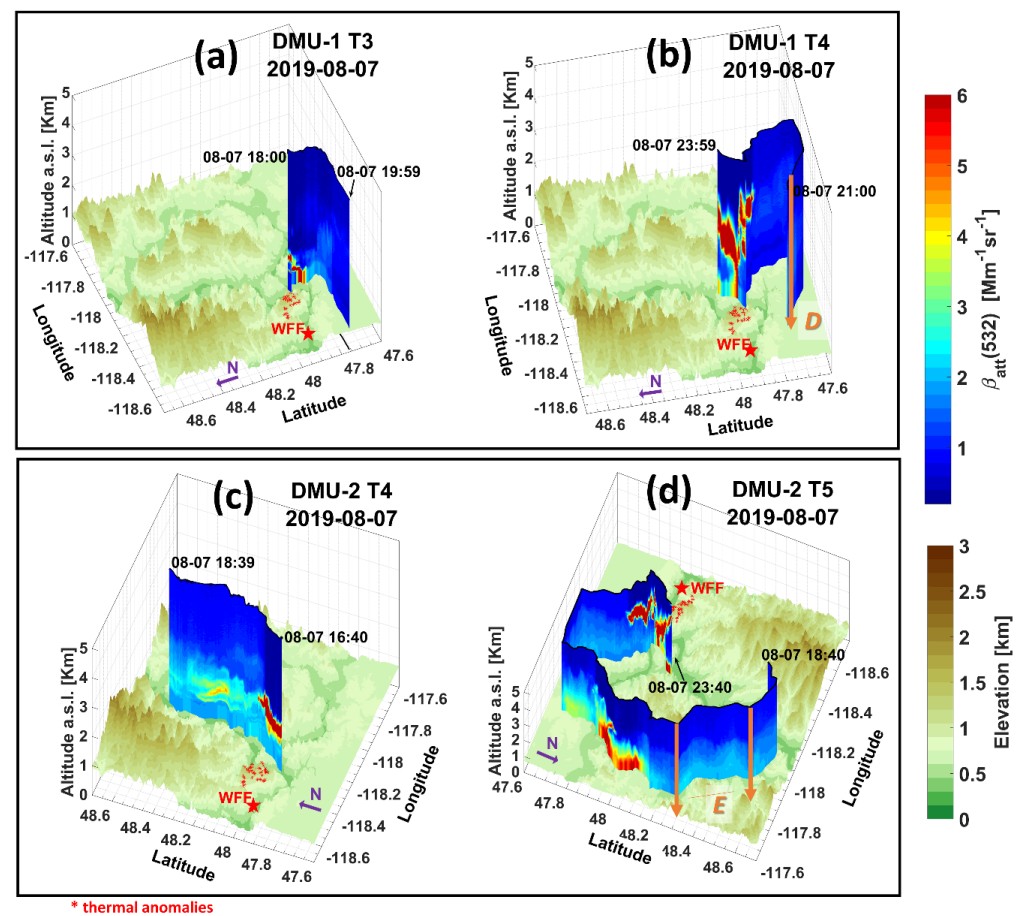

**Figure 11** Spatial-temporal distribution of total attenuated backscatter at 532 nm, same as Fig. 10 but for the trajectories during 2019-08-07.




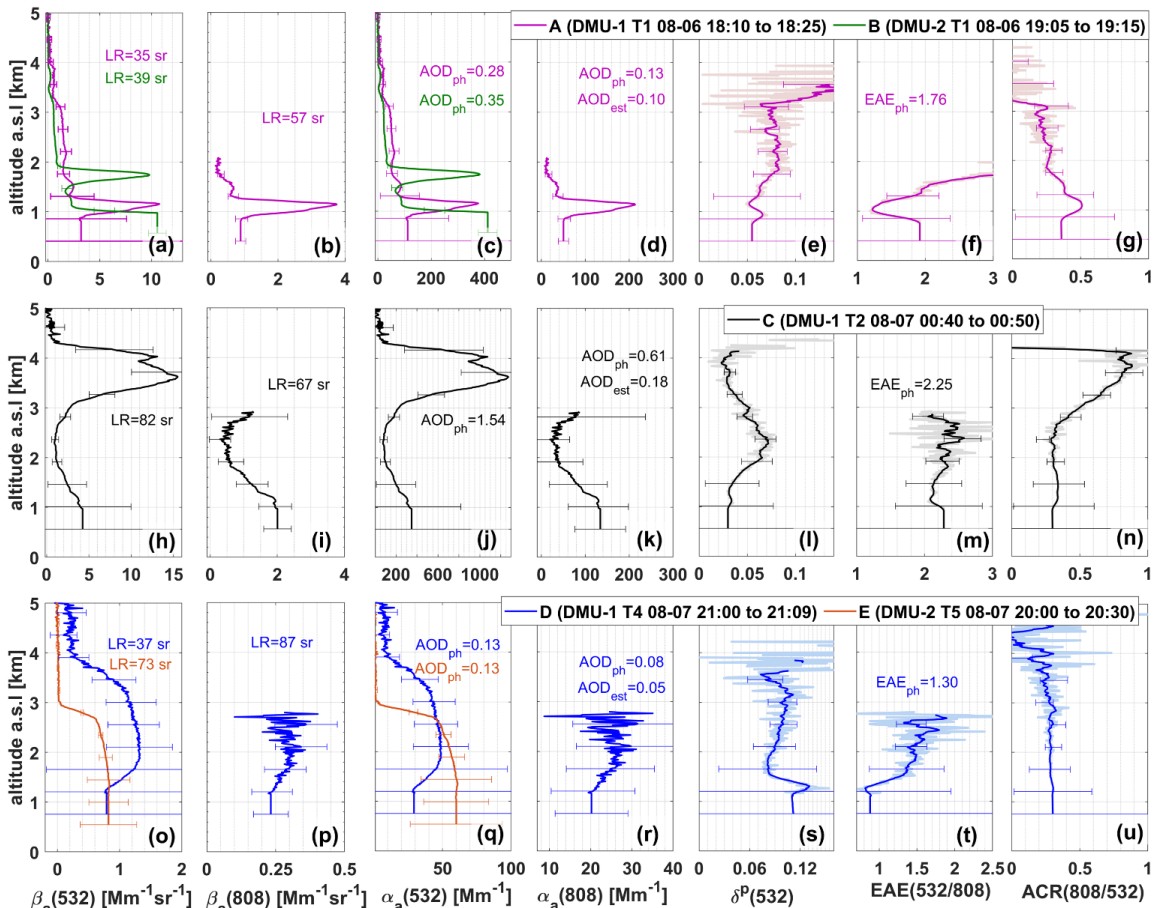

**Figure 12**. Profiles of aerosol optical properties from averaged selected datasets of both DMU-1 and DMU-2 mobile observations during 2019-08-06 and 2019-08-07. The selected data is displayed in Fig. 10 and Fig. 11 by orange arrows on the 3D $\beta_{att}$ distributions. Each dataset is differentiated by color. Profiles of backscatter at 532 nm (a, h, o) and 808 nm (b, i, p), extinction at 532 nm (c, j, q) and 808 nm (d, k, r), PLDR (e, l, s), EAE (f, m, t) and ACR (g, n, u).