# Peer review of "Enhancing Mobile Aerosol Monitoring with CE376 Dual-Wavelength Depolarization Lidar"

_EGUsphere, 2023_

## Author Comment (AC1)

We would like to thank the two anonymous referees for their insightful comments and constructive feedback, that are highly appreciated. Your thorough review helps us to refine our arguments and strengthen our conclusions. Please find below our responses. The referees' comments (RCs) are listed below in black and the authors' answers are listed in blue. The figures added within the responses are named as Figure R.

**Modification in the figures**

We noticed that the error associated to EAE profiles were smaller than expected. This due to an error of sign in the equation inside the algorithm, which have been corrected. All the figures that included EAE profiles were updated with the correct equation for the error estimation. Now, the error bars for EAE profiles are larger by a factor of  $\sim$ 4.

**Anonymous referee #1**

The micropulse lidars (MPL) are popular tools for the study of aerosol, due to their relatively low price, eye safety and capability for continuous operation. However, in these lidars only elastic backscatter is analyzed, which poses the complications for retrieval the aerosol backscattering and extinction coefficients. Thus, the novel approaches for the MPL data handling are welcome by the lidar community. The manuscript provides description of the new dual wavelength MPL Cimel CE376 as well as approach to data analysis, based on the use the AOD measured by the Sun Photometer. It is important, that authors performed MPL observations collocated with Mie-Raman lidar, allowing to validate their approach. The mobile lidar observations at the proximity of fires also demonstrate new interesting results. The manuscript is well written, contains new useful information and suitable for AMT.

**Technical comments:**

1. I would suggest to compare lidar ratios used in calculations with values provided by AERONET.

We consider this point really important. However, lidar ratios from AERONET are provided when the single scattering albedo is available, i.e., when sky radiance and direct solar measurements are available and when AOD(440) is greater or equal to 0.4 for quality assured data level 2 (Holben et al., 2006). In this context, the case study of Saharan dust in spring 2021 (Sect. 4.2.1) is the only case that we can use for comparisons with AERONET lidar ratios. For the other case study at ATOLL (dust and smoke, Sect. 4.2.2), AOD values are lower than 0.4. More details about this point are discussed in the point 7.

2. Ln 123. Would be good to have more information about 808 nm diode: linewidth, pulse duration, manufacturer.

We added this information in the text. For instance, the diode laser manufacturer was DILAS (now manufactured by Coherent). The 808 nm diode linewidth is 0.4 nm, central wavelength is 808 nm and pulse width of 186 ns.

**3. Ln 135-139. The system uses grid polarizers, so no reason to discuss crosstalk between channels.**

We agree, the discussion on crosstalk between channels is not necessary when the system uses grid polarizers. However, we recall that the case studies included in this work are at different operational conditions, i.e., with (Sect. 4.2) and without (Sect. 5) grid polarizers. Thus, we consider convenient to mention the PBS transmittivities and reflectivities in the discussion of results to maintain clarity and we modified the text in Ln 135-139 to simplify the text. Please find below the modified text.

Linear depolarization measurements at 532 nm are also acquired by separation in parallel (copolarized) and perpendicular (cross-polarized) components of the backscattered light using a polarizing beamsplitter cube (PBS) in the reception. The PBS is a Thorlabs CCM1-PBS25-532 with reflectivities Rp and Rs and transmittances Tp and Ts (subscripts p and s for parallel and perpendicular polarized light with respect to the PBS incident plane). A manually-rotating mount with half-wave plate (HWP) in front of the PBS controls the polarization angle of the incident light with a precision of 2 degrees. Measured signals behind the PBS on the reflected and transmitted branches are named parallel (//) or perpendicular ( $\perp$ ) according to the reception configuration. More details on the depolarization measurements can be found in Sect. 3.1.1.

**4. Ln 230-235. It was published many times, so probably no reason to repeat.**

We agree, we removed the equations to derive the total RCS and VLDR. The simplified section is presented in the following lines.

**3.1.1 Volume Linear Depolarization Ratio**

The total RCS and VLDR,  $\delta^{\nu}(r)$ , at 532 nm are derived following the methods described by Freudenthaler et al. (2009). Rotating the HWP, the angle  $\varphi$  between the plane of polarization of the laser and the incident plane of the PBS can be changed for two arrangements ( $\varphi=0^{\circ}$  or  $90^{\circ}$ ). For commercial PBS cubes (Rs>Rp and Tp>Ts), the system configuration at  $\varphi=0^{\circ}$  is defined when the parallel polarized signal is measured in the transmitted branch of the PBS. Moreover, to reduce noise and errors from cross-talk effects, the configuration  $\varphi=90^{\circ}$  can be also considered. The relative amplification factor V\* is calculated using the ±45° calibration (Freudenthaler et al., 2009), under cloud free and stable atmospheric conditions.

The HWP rotates the angle of the incident polarization plane  $\varphi$  by means of 2 $\theta$  with  $\theta$  precision of 2°. The error induced by the uncertainty in  $\varphi$  represent less than 5% of error on V\* for VLDR values up to 0.3 (Figure 2, Freudenthaler et al., 2009). Moreover, to improve depolarization measurements, wiregrid polarizers can be added to the PBS to reduce the cross-talk. However, additional errors during the calibration and in regular measurements can come from polarizing optical components that need detailed characterization (Freudenthaler, 2016), which are not considered in this work. For current versions of the CE376, a motorized PBS mount is integrated.

5. Section 3.1.3. I am a little confused. Authors introduce EAE for 532-808 nm. But for Klett method, extinction and backscattering profiles have the same shape (we can see it in Fig.5). Does CR bring new information comparing to EAE?

Indeed, CR doesn't bring new information compared to EAE, both are derived using vertically constant LR. We mainly wanted to emphasize the new product ACR, which is directly obtained from the total attenuated backscatter signals, the CR is therefore used only for comparison with ACR.

6. Ln 298. "we retrieve LR, extinction, backscattering at both wavelengths". I would not call it retrieval, because there are 4 unknowns and 2 equations. It is more like estimation.

Yes, this is true, we modified the text accordingly.

7. Fig.5f. Lidar ratio at 808 nm is higher than at 532 nm, which looks unusual for me. Did authors compare their results with lidar ratios provided by AERONET? LILAS LR above 4 km is probably untrustable.

The profiles presented in Fig 5 correspond to night-time measurements (20:00 to 22:00 UT, 1 April 2021), so it was not possible, for the same time threshold, to compare against lidar ratios from AERONET. However, the AERONET lidar ratio values during day time, on the same day (1 April 2021),

are on average 56 sr, 55 sr and 61 sr at 440 nm, 870 nm and 1020 nm respectively (data level 2). Towards 17:00 UT on 1 April 2021, the LR values increased (Fig. R.1) up to 62 at 440 nm and 870 nm, and up to 71 sr at 1020 nm. The LR values from photometer for the wavelength pair 440-870 nm do not present spectral variation like for 440 nm and 1020 nm. Nevertheless, the last calculated set of LR from photometer at ~17:00 UT showed increasing LR with larger wavelength (57 sr at 440 nm, 60 sr at 870 nm and 69 sr at 1020 nm). Moreover, the AOD and EAE from sun/moon photometer measurements (Fig. 3c) showed that the values before and after the time considered for the profiles in Fig. 5, changed drastically, i.e., high AOD (>0.8) and low EAE(<0.5) values before, and low AOD (<0.25) and high EAE (>0.7) after. Thus, it is expected a temporal variation in the optical properties, as suggested by the observations, higher LR values at longer wavelengths. Furthermore, the LR obtained with METIS, 54  $\pm$  3 sr at 532 nm and 69  $\pm$  4 sr at 808 nm (Fig. R.1), are comparable to values presented in a previous work by Haarig et al. (2022), 50  $\pm$  5 sr at 532 nm and 69  $\pm$  14 sr at 1064 nm, which used Raman method at both wavelengths during a similar dust event on February 2021 in Germany.

Figure R.1: Comparison of LR obtained from AERONET (LRph) and LR obtained from METIS (LRlid).

We agree, we corrected in the text.

**Anonymous referee #2**

This study aims to demonstrate the capabilities of a compact dual-wavelength depolarisation lidar in evaluating the spatiotemporal distribution of aerosol properties, especially when deployed on mobile platforms and collocated with a photometer. A modified Klett inversion method was employed. Firstly, some results were compared with a reliable Mie-Raman lidar under different aerosol scenarios, to evaluate both the method's efficacy and the instrument's suitability. Afterwards, the system was employed during the FIREX-AQ campaign in the summer 2019.

The manuscript is well-written and structured, and the insights are novel and valuable. It is appropriate for publication in AMT.

**Minor Revisions**

1. Lines 37-38: I suggest adding also the reference Papagiannopoulos et al. (2020): An EARLINET early warning system for atmospheric aerosol aviation hazards, Atmos. Chem. Phys., 20, 10775–10789, https://doi.org/10.5194/acp-20-10775-2020, 2020.

We added the reference in the text.

 Line 40: I suggest adding also the reference Pappalardo et al., EARLINET: towards an advanced sustainable European aerosol lidar network, Atmos. Meas. Tech., 7, 2389-2409, doi:10.5194/amt-7-2389-2014, 2014.

We added the reference in the text.

3. Lines 59-61: This statement is generally true. However, there are some exceptions. For instance, lidars from MPLNET and PollyXT (Engelmann et al., 2016; doi:10.5194/amt-9-1767-2016) are compact and easily transportable lidars, capable of operating 24/7 automatically. Please, qualify that statement.

We agree, we developed more the statement in the text. Please find below the modified text.

Lidar systems are mostly large, complex, require considerable space, regular maintenance and controlled operational conditions. Upgrades for mobile applications are frequently linked to instrumental modifications and/or creation of adapted laboratory platforms or transportable containers. Examples are the multiwavelength PollyXT lidars, within the network PollyNET (Althausen et al., 2013; Engelman et al., 2016) set up in temperature-controlled containers for 24/7 operation, and the micro-pulse lidars from MPLNET, which are automatic, compact systems that can be easily transported.

4. Line 121: could you specify what the acronyms 'G, GP, GPN, N' mean?

The acronyms G (Green), GP (Green Polarized), GPN (Green Polarized Near-infrared) and N (Near-infrared) state the possible configurations of the CE376 lidar. We included the acronyms in the text.

5. Lines 148-151: the authors explained the different levels of data from the photometers used in this study. Please specify in the text which level of data (LV1.5 or LV2.0) has been employed.

We added in the text the AERONET data level used in the studies. Also, more details concerning the photometer data processing have been included. Please find below the modified paragraph.

The CIMEL CE318-T photometer has been adapted for mobile applications. The PLASMA photometer has been developed exclusively for mobile observations. Both instruments follow and meet the AERONET standards and are included in automatic data processing chains. Therefore, automatic near

real time (NRT) aerosol properties are retrieved (https://aeronet.gsfc.nasa.gov, last access: 23 October 2023), without cloud screening as data level 1.0, and with cloud screening as data level 1.5. It is important to note that AERONET cloud screening was formulated for stationary instruments and some additional uncertainty in the cloud screening technique may either identify thin clouds as aerosols or vice versa, especially in the presence of smoke or dust plumes. Further, cirrus cloud screening employed by AERONET Version 3 may be further limited (Giles et al., 2019). After calibration, quality assured data at level 2.0 is also acquired (Smirnov et al., 2000, Giles et al., 2019). In this work, data level 2.0 is used for stationary measurements (Sect. 4) and data level 1.5 is used for mobile measurements (Sect. 5). Both photometers are used in this work and are briefly described below.

 Lines 212-213: I suggest adding also the reference Córdoba-Jabonero et al. (2021): Experimental assessment of a micro-pulse lidar system in comparison with reference lidar measurements for aerosol optical properties retrieval, Atmos. Meas. Tech, 14, 5225-5239, doi:10.5194/amt-14-5225-2021.

**We included the reference.**

7. Line 370 (section 4.2.1): the authors investigate a case of dust over Lille. Indeed, the optical properties are characteristics of this type of aerosol. However, any ancillary analysis supports the assertion that it is Saharan dust. For instance, it should be interesting to include an analysis that confirms the source of the dust. For instance, back trajectories analysis from HYSPLIT model (https://www.ready.noaa.gov/hypub-bin/trajtype.pl)

We agree that back-trajectories analysis is important to confirm the aerosol sources. Indeed, we took them into consideration for the analysis of both case studies at ATOLL, Sect. 4.2.1 and Sect. 4.2.2, nevertheless we didn't include them in the text due to the extent of the publication. We added a reference to the analysis without expanding the text. Please find the ancillary analysis made for each case in the following lines.

**Ancillary analysis for case studies at ATOLL (Lille, France)**

**A.1 Saharan dust transport over Lille (31 March to 2 April 2021)**

Figure R.2 presents total column Dust Optical Depth (DOD) at 550 nm showing the predictions of dust transport from North-Western Africa towards Western Europe. The images correspond to the period when the dust event was observed at ATOLL: 31 March 2021 at 12:00 (Fig. R.2a), 01 April 2021 at 00:00 (Fig. R.2b), 01 April 2021 at 12:00 (Fig. R.2c) and 02 April 2021 at 12:00 (Fig. R.2d). For DOD images, the multi-model forecast, mean values over 11 models, is considered (Basart et al., 2019) which is provided by the WMO Barcelona Dust Regional Center and the partners of the Sand and Dust Storm Warning Advisory and Assessment System (SDS-WAS) for Northern Africa, the Middle East and Europe (https://dust.aemet.es/products/daily-dust-products). The forecast shows the dust airmasses passing over Portugal-Spain towards the Northern Atlantic and reaching UK and Northern France with DOD(550) values up to 0.8. According to the event development, in particular, ATOLL is impacted by the highest DOD values during 1 April 2021.

The NOAA HYSPLIT back trajectory analysis (Stein et al., 2015) is used to identify the possible sources of the transported aerosols. In this case, the GFSQ (Global Forecast System 0.25 degrees) dataset is considered as the meteorological input for the model and the vertical velocity of airmass is modeled using vertical motion velocity calculation; the model is run for 5 days of transport. Figure R.3 presents back-trajectories ending at 3 altitudes above ATOLL platform (500 m, 2 km and 4 km) on 1 April 2021 at 00:00 UT (Fig. R.3a), 12:00 UT (Fig. R.3b) and 21:00 UT (Fig. R.3c). These back-trajectories confirm that the airmasses between 1.5-6 km asl follow the paths of transported dust as forecasted by the DOD

images (Fig. R.2). Contrary, at lower altitudes within the ABL, back-trajectories suggest different sources along the day probably influenced by urban/industrial emissions (Fig. R.3a and Fig. R.3b) as reported for ATOLL by previous studies (Velazquez Garcia, 2023; Velazquez-Garcia et al., 2023). In this context, the lidar observations showing two distinct aerosol contributions, below and above the ABL, is supported.

---

## Author Comment (AC2)

We would like to thank the two anonymous referees for their insightful comments and constructive feedback, that are highly appreciated. Your thorough review helps us to refine our arguments and strengthen our conclusions. Please find below our responses. The referees' comments (RCs) are listed below in black and the authors' answers are listed in blue. The figures added within the responses are named as Figure R.

**Modification in the figures**

We noticed that the error associated to EAE profiles were smaller than expected. This due to an error of sign in the equation inside the algorithm, which have been corrected. All the figures that included EAE profiles were updated with the correct equation for the error estimation. Now, the error bars for EAE profiles are larger by a factor of ~4.

**Anonymous referee #1**

The micropulse lidars (MPL) are popular tools for the study of aerosol, due to their relatively low price, eye safety and capability for continuous operation. However, in these lidars only elastic backscatter is analyzed, which poses the complications for retrieval the aerosol backscattering and extinction coefficients. Thus, the novel approaches for the MPL data handling are welcome by the lidar community. The manuscript provides description of the new dual wavelength MPL Cimel CE376 as well as approach to data analysis, based on the use the AOD measured by the Sun Photometer. It is important, that authors performed MPL observations collocated with Mie-Raman lidar, allowing to validate their approach. The mobile lidar observations at the proximity of fires also demonstrate new interesting results. The manuscript is well written, contains new useful information and suitable for AMT.

**Technical comments:**
1. I would suggest to compare lidar ratios used in calculations with values provided by AERONET.

We consider this point really important. However, lidar ratios from AERONET are provided when the single scattering albedo is available, i.e., when sky radiance and direct solar measurements are available and when AOD(440) is greater or equal to 0.4 for quality assured data level 2 (Holben et al., 2006). In this context, the case study of Saharan dust in spring 2021 (Sect. 4.2.1) is the only case that we can use for comparisons with AERONET lidar ratios. For the other case study at ATOLL (dust and smoke, Sect. 4.2.2), AOD values are lower than 0.4. More details about this point are discussed in the point 7.

2. Ln 123. Would be good to have more information about 808 nm diode: linewidth, pulse duration, manufacturer.

We added this information in the text. For instance, the diode laser manufacturer was DILAS (now manufactured by Coherent). The 808 nm diode linewidth is 0.4 nm, central wavelength is 808 nm and pulse width of 186 ns.

3. Ln 135-139. The system uses grid polarizers, so no reason to discuss crosstalk between channels.

We agree, the discussion on crosstalk between channels is not necessary when the system uses grid polarizers. However, we recall that the case studies included in this work are at different operational conditions, i.e., with (Sect. 4.2) and without (Sect. 5) grid polarizers. Thus, we consider convenient to mention the PBS transmittivities and reflectivities in the discussion of results to maintain clarity and we modified the text in Ln 135-139 to simplify the text. Please find below the modified text.

*Linear depolarization measurements at 532 nm are also acquired by separation in parallel (co-polarized) and perpendicular (cross-polarized) components of the backscattered light using a polarizing*

*beamsplitter cube (PBS) in the reception. The PBS is a Thorlabs CCM1-PBS25-532 with reflectivities Rp and Rs and transmittances Tp and Ts (subscripts p and s for parallel and perpendicular polarized light with respect to the PBS incident plane). A manually-rotating mount with half-wave plate (HWP) in front of the PBS controls the polarization angle of the incident light with a precision of 2 degrees. Measured signals behind the PBS on the reflected and transmitted branches are named parallel (//) or perpendicular (⊥) according to the reception configuration. More details on the depolarization measurements can be found in Sect. 3.1.1.*

4. Ln 230-235. It was published many times, so probably no reason to repeat.

We agree, we removed the equations to derive the total RCS and VLDR. The simplified section is presented in the following lines.

**3.1.1 Volume Linear Depolarization Ratio**

*The total RCS and VLDR, $\delta^v(r)$, at 532 nm are derived following the methods described by Freudenthaler et al. (2009). Rotating the HWP, the angle φ between the plane of polarization of the laser and the incident plane of the PBS can be changed for two arrangements (φ=0° or 90°). For commercial PBS cubes (Rs>Rp and Tp>Ts), the system configuration at φ=0° is defined when the parallel polarized signal is measured in the transmitted branch of the PBS. Moreover, to reduce noise and errors from cross-talk effects, the configuration φ=90° can be also considered. The relative amplification factor V\* is calculated using the ±45° calibration (Freudenthaler et al., 2009), under cloud free and stable atmospheric conditions.*

*The HWP rotates the angle of the incident polarization plane φ by means of 2θ with θ precision of 2°. The error induced by the uncertainty in φ represent less than 5% of error on V\* for VLDR values up to 0.3 (Figure 2, Freudenthaler et al., 2009). Moreover, to improve depolarization measurements, wire-grid polarizers can be added to the PBS to reduce the cross-talk. However, additional errors during the calibration and in regular measurements can come from polarizing optical components that need detailed characterization (Freudenthaler, 2016), which are not considered in this work. For current versions of the CE376, a motorized PBS mount is integrated.*

5. Section 3.1.3. I am a little confused. Authors introduce EAE for 532-808 nm. But for Klett method, extinction and backscattering profiles have the same shape (we can see it in Fig.5). Does CR bring new information comparing to EAE?

Indeed, CR doesn't bring new information compared to EAE, both are derived using vertically constant LR. We mainly wanted to emphasize the new product ACR, which is directly obtained from the total attenuated backscatter signals, the CR is therefore used only for comparison with ACR.

6. Ln 298. "we retrieve LR, extinction, backscattering at both wavelengths". I would not call it retrieval, because there are 4 unknowns and 2 equations. It is more like estimation.

Yes, this is true, we modified the text accordingly.

7. Fig.5f. Lidar ratio at 808 nm is higher than at 532 nm, which looks unusual for me. Did authors compare their results with lidar ratios provided by AERONET? LILAS LR above 4 km is probably untrustable.

The profiles presented in Fig 5 correspond to night-time measurements (20:00 to 22:00 UT, 1 April 2021), so it was not possible, for the same time threshold, to compare against lidar ratios from AERONET. However, the AERONET lidar ratio values during day time, on the same day (1 April 2021),

are on average 56 sr, 55 sr and 61 sr at 440 nm, 870 nm and 1020 nm respectively (data level 2). Towards 17:00 UT on 1 April 2021, the LR values increased (Fig. R.1) up to 62 at 440 nm and 870 nm, and up to 71 sr at 1020 nm. The LR values from photometer for the wavelength pair 440-870 nm do not present spectral variation like for 440 nm and 1020 nm. Nevertheless, the last calculated set of LR from photometer at ~17:00 UT showed increasing LR with larger wavelength (57 sr at 440 nm, 60 sr at 870 nm and 69 sr at 1020 nm). Moreover, the AOD and EAE from sun/moon photometer measurements (Fig. 3c) showed that the values before and after the time considered for the profiles in Fig. 5, changed drastically, i.e., high AOD (>0.8) and low EAE(<0.5) values before, and low AOD (<0.25) and high EAE (>0.7) after. Thus, it is expected a temporal variation in the optical properties, as suggested by the observations, higher LR values at longer wavelengths. Furthermore, the LR obtained with METIS, $54 \pm 3$ sr at 532 nm and $69 \pm 4$ sr at 808 nm (Fig. R.1), are comparable to values presented in a previous work by Haarig et al. (2022), $50 \pm 5$ sr at 532 nm and $69 \pm 14$ sr at 1064 nm, which used Raman method at both wavelengths during a similar dust event on February 2021 in Germany.

[Figure]

**Figure R.1:** Comparison of LR obtained from AERONET ($LR_{ph}$) and LR obtained from METIS ($LR_{lid}$).

8. Ln 406. VLDR provides not much information for comparison. So better focus at PLDR.

We agree, we corrected in the text.

**Anonymous referee #2**

This study aims to demonstrate the capabilities of a compact dual-wavelength depolarisation lidar in evaluating the spatiotemporal distribution of aerosol properties, especially when deployed on mobile platforms and collocated with a photometer. A modified Klett inversion method was employed. Firstly, some results were compared with a reliable Mie-Raman lidar under different aerosol scenarios, to evaluate both the method's efficacy and the instrument's suitability. Afterwards, the system was employed during the FIREX-AQ campaign in the summer 2019.

The manuscript is well-written and structured, and the insights are novel and valuable. It is appropriate for publication in AMT.

**Minor Revisions**

1. Lines 37-38: I suggest adding also the reference Papagiannopoulos et al. (2020): An EARLINET early warning system for atmospheric aerosol aviation hazards, Atmos. Chem. Phys., 20, 10775–10789, https://doi.org/10.5194/acp-20-10775-2020, 2020.

We added the reference in the text.

2. Line 40: I suggest adding also the reference Pappalardo et al., EARLINET: towards an advanced sustainable European aerosol lidar network, Atmos. Meas. Tech., 7, 2389-2409, doi:10.5194/amt-7-2389-2014, 2014.

We added the reference in the text.

3. Lines 59-61: This statement is generally true. However, there are some exceptions. For instance, lidars from MPLNET and PollyXT (Engelmann et al., 2016; doi:10.5194/amt-9-1767-2016) are compact and easily transportable lidars, capable of operating 24/7 automatically. Please, qualify that statement.

We agree, we developed more the statement in the text. Please find below the modified text.

*Lidar systems are mostly large, complex, require considerable space, regular maintenance and controlled operational conditions. Upgrades for mobile applications are frequently linked to instrumental modifications and/or creation of adapted laboratory platforms or transportable containers. Examples are the multiwavelength Polly$^{XT}$ lidars, within the network PollyNET (Althausen et al., 2013; Engelman et al., 2016) set up in temperature-controlled containers for 24/7 operation, and the micro-pulse lidars from MPLNET, which are automatic, compact systems that can be easily transported.*

4. Line 121: could you specify what the acronyms 'G, GP, GPN, N' mean?

The acronyms G (Green), GP (Green Polarized), GPN (Green Polarized Near-infrared) and N (Near-infrared) state the possible configurations of the CE376 lidar. We included the acronyms in the text.

5. Lines 148-151: the authors explained the different levels of data from the photometers used in this study. Please specify in the text which level of data (LV1.5 or LV2.0) has been employed.

We added in the text the AERONET data level used in the studies. Also, more details concerning the photometer data processing have been included. Please find below the modified paragraph.

*The CIMEL CE318-T photometer has been adapted for mobile applications. The PLASMA photometer has been developed exclusively for mobile observations. Both instruments follow and meet the AERONET standards and are included in automatic data processing chains. Therefore, automatic near*

*real time (NRT) aerosol properties are retrieved (https://aeronet.gsfc.nasa.gov, last access: 23 October 2023), without cloud screening as data level 1.0, and with cloud screening as data level 1.5. It is important to note that AERONET cloud screening was formulated for stationary instruments and some additional uncertainty in the cloud screening technique may either identify thin clouds as aerosols or vice versa, especially in the presence of smoke or dust plumes. Further, cirrus cloud screening employed by AERONET Version 3 may be further limited (Giles et al., 2019). After calibration, quality assured data at level 2.0 is also acquired (Smirnov et al., 2000, Giles et al., 2019). In this work, data level 2.0 is used for stationary measurements (Sect. 4) and data level 1.5 is used for mobile measurements (Sect. 5). Both photometers are used in this work and are briefly described below.*

6. Lines 212-213: I suggest adding also the reference Córdoba-Jabonero et al. (2021): Experimental assessment of a micro-pulse lidar system in comparison with reference lidar measurements for aerosol optical properties retrieval, Atmos. Meas. Tech, 14, 5225-5239, doi:10.5194/amt-14-5225-2021.

We included the reference.

7. Line 370 (section 4.2.1): the authors investigate a case of dust over Lille. Indeed, the optical properties are characteristics of this type of aerosol. However, any ancillary analysis supports the assertion that it is Saharan dust. For instance, it should be interesting to include an analysis that confirms the source of the dust. For instance, back trajectories analysis from HYSPLIT model (https://www.ready.noaa.gov/hypub-bin/trajtype.pl)

We agree that back-trajectories analysis is important to confirm the aerosol sources. Indeed, we took them into consideration for the analysis of both case studies at ATOLL, Sect. 4.2.1 and Sect. 4.2.2, nevertheless we didn't include them in the text due to the extent of the publication. We added a reference to the analysis without expanding the text. Please find the ancillary analysis made for each case in the following lines.

**Ancillary analysis for case studies at ATOLL (Lille, France)**

**A.1 Saharan dust transport over Lille (31 March to 2 April 2021)**

Figure R.2 presents total column Dust Optical Depth (DOD) at 550 nm showing the predictions of dust transport from North-Western Africa towards Western Europe. The images correspond to the period when the dust event was observed at ATOLL: 31 March 2021 at 12:00 (Fig. R.2a), 01 April 2021 at 00:00 (Fig. R.2b), 01 April 2021 at 12:00 (Fig. R.2c) and 02 April 2021 at 12:00 (Fig. R.2d). For DOD images, the multi-model forecast, mean values over 11 models, is considered (Basart et al., 2019) which is provided by the WMO Barcelona Dust Regional Center and the partners of the Sand and Dust Storm Warning Advisory and Assessment System (SDS-WAS) for Northern Africa, the Middle East and Europe (https://dust.aemet.es/products/daily-dust-products). The forecast shows the dust airmasses passing over Portugal-Spain towards the Northern Atlantic and reaching UK and Northern France with DOD(550) values up to 0.8. According to the event development, in particular, ATOLL is impacted by the highest DOD values during 1 April 2021.

The NOAA HYSPLIT back trajectory analysis (Stein et al., 2015) is used to identify the possible sources of the transported aerosols. In this case, the GFSQ (Global Forecast System 0.25 degrees) dataset is considered as the meteorological input for the model and the vertical velocity of airmass is modeled using vertical motion velocity calculation; the model is run for 5 days of transport. Figure R.3 presents back-trajectories ending at 3 altitudes above ATOLL platform (500 m, 2 km and 4 km) on 1 April 2021 at 00:00 UT (Fig. R.3a), 12:00 UT (Fig. R.3b) and 21:00 UT (Fig. R.3c). These back-trajectories confirm that the airmasses between 1.5-6 km asl follow the paths of transported dust as forecasted by the DOD

images (Fig. R.2). Contrary, at lower altitudes within the ABL, back-trajectories suggest different sources along the day probably influenced by urban/industrial emissions (Fig. R.3a and Fig. R.3b) as reported for ATOLL by previous studies (Velazquez Garcia, 2023; Velazquez-Garcia et al., 2023). In this context, the lidar observations showing two distinct aerosol contributions, below and above the ABL, is supported.

[Figure]

**Figure R.2:** Forecast images of total column dust optical depth at 550 nm over Northern Africa, Middle East and Europe. (a) 31 March 2021 at 12:00, (b) 01 April 2021 at 00:00, (c) 01 April 2021 at 12:00 and (d) 02 April 2021 at 12:00. Images are generated with multi-model forecast and are provided by WMO Barcelona Dust Regional Center and the partners of SDS-WAS (https://dust.aemet.es/products/daily-dust-products).

[Figure]

**Figure R.3:** NOAA HYSPLIT back-trajectories ending at 500 m, 2 km and 4 km above ATOLL (Lille, France). Different ending hours during 1 April 2021 are considered: (a) 00:00, (b) 12:00 and (c) 21:00 UT. GFSQ meteorological data is used to run the model for 5 days of transport. The web interface was used to obtain these results (https://www.ready.noaa.gov/HYSPLIT_traj.php).

**A.2 Saharan dust and Smoke transport over Lille (17 to 20 July 2022)**

DOD(550) images from 15 July 2022 to 19 July 2022 (Fig R.4a to Fig. R.4e) show the evolution of the dust transport path from the North Western Sahara passing through Portugal-Spain passing above the Atlantic towards France. The images correspond to the model run at 12:00 for each day. The active fires are also indicated on top MODIS image for the 15 July 2022 (Fig. R.4f), and are located on the predicted path of dust for that day. Moreover, according to the dust transport path, ATOLL site was most likely to

be influenced by DOD(550) values between 0.1 and 0.2 during 17 and 18 July 2022. Therefore, at the time that the heatwave traversed Lille, both dust and smoke were detected in the atmospheric column.

[Figure]

**Figure R.4:** Forecast images of total column dust optical depth at 550 nm over Northern Africa, Middle East and Europe, and satellite image showing active fires over Western Europe. Forecast images correspond to (a) 15 July 2022 at 12:00, (b) 16 July 2022 at 12:00, (c) 17 July 2022 at 12:00, (d) 18 July 2022 at 12:00 and (e) 19 July 2022 at 12:00. (f) MODIS-AQUA True Color Image for 15 July 2022 over Western Europe is also presented, where the thermal anomalies (active fires) are indicated by orange dots and a red flame for Gironde Fires. Forecast images are generated with multi-model forecast and are provided by WMO Barcelona Dust Regional Center and the partners of SDS-WAS (https://dust.aemet.es/products/daily-dust-products). MODIS products are available through NASA Worldview Snapshots (https://wvs.earthdata.nasa.gov).

[Figure]

**Figure R.5:** NOAA HYSPLIT back-trajectories ending at 1.5 km (a) and 4 km (b) above ATOLL (Lille, France). Each trajectory corresponds to a different ending hour during 17 July 2022, every 3 h since 09:00 UT. GFSQ meteorological data is used to run the model for 5 days of transport. The web interface was used to obtain these resutls (https://www.ready.noaa.gov/HYSPLIT_traj.php).

*During 17 July 2022*, a layer extended between ~3 km to 5 km asl is lofted on top of apparently clean air, as indicated by the low values on VLDR and extinction height-temporal variations (Fig. 6 panels b, c and d). Likewise, HYSPLIT back-trajectories ending at 1.5 km and 4 km above ATOLL corroborate

that different airmasses are impacting the site (Fig. R.5). The transport path of the aerosol layer detected at ~4 km asl (Fig. R.5b) coincide with the dust transport track forecasted on the DOD(550) images, i.e., crossing the Portugal-Spain fires (Fig. R.4). Moreover, the cleaner air intrusion (~1.5 km asl) is influenced by airmasses transported from higher altitudes above the North Atlantic Ocean (Fig. R.5a). The back-trajectories are modeled for 3 days of transport with GFSQ meteorological database.

*During the second period of 18-19 July 2022*, the layer from the day before now reduced to ~0.5 km width is descending from 3 km towards 1.5 km asl, accompanied by 2 separated layers above it. In particular, we focus our attention on the afternoon of 18 July 2022 to early morning of 19 July 2022, where quite stable $AOD_{ph}$ and $EAE_{ph}$ are observed. Back-trajectories ending at 3 altitudes, 1.7 km (L1 in red), 2.7 km (L2 in blue) and 4 km (L3 in green) above ATOLL, and at different arrival hours (model run for 5.5 days) are presented in Fig. R.6.

[Figure]

**Figure R.6:** NOAA HYSPLIT back-trajectories ending at 1.8 km, 2.7 km and 4 km above ATOLL (Lille, France). Different arrival times are considered, 18:00 UT (a) and 21:00 UT (b) on 18 July 2022, 00:00 UT (c) and 03:00 UT (d) on 19 July 2022. GFSQ meteorological data is used to run the model for 5 days of transport. The web interface was used to obtain these resutls (https://www.ready.noaa.gov/HYSPLIT_traj.php).

From the back-trajectories, similarly to the day before (17 July 2022) is observed, the layer between 3.2 to 4.5 km asl (L3 in green) follows the path of *dust crossing Portugal fires* as predicted by DOD images (Fig. R.4). In contrast, back-trajectories for the layer between 2.4 km and 3.2 km asl (L2 in blue), with lower VLDR values and higher extinction (Fig. 6), show that airmasses follow a quite different path than the transported dust. This time the source is identified at South West France and North East Spain, closer to the Gironde region where active fires were detected. On the other hand, L1 in red show that airmasses are transported over or near the dust transport path and pass over active fires. Therefore, the layer L2 appears to be under higher influence of smoke aerosols rather than dust like L1 and L3.

**Technical Revisions**

8.   Line 72: The acronym LILAS is not defined until line 346. Please, correct it.
We corrected it in the text.

9. Line 165: Please, specify what 'MAP-IO' stands for.
We added the acronym explanation in the text.
10. Line 194: Please, correct the format of the reference Kovalev and William (2004).
We corrected the reference in the text.
11. Lines 319-320: Please, order the references by year.
We corrected the reference in the text.
12. Line 401: The text refers to Fig. 5f, not to Fig. 4f.
We corrected the reference in the text.

**References**

Althausen, D., Engelmann, E., Baars, H., Heese, B., Kanitz, T., Komppula, M., Giannakaki, E., Pfüller, A., Silva, A. M., Preißler, J., Wagner, F., Rascado, J. L., Pereira, S., Lim, J. H., Ahn, J. Y., Tesche, M., and Stachlewska, I. S.: PollyNET – a network of multiwavelength polarization Raman lidars, in: Proc. SPIE 8894, Lidar Technologies, Techniques, and Measurements for Atmospheric Remote Sensing IX, 88940I, 22 October 2013, Dresden, Germany, 8894, doi:10.1117/12.2028921, 2013.

Basart, S., Nickovic, S., Terradellas, E., Cuevas, E., Pérez García-Pando, C., García-Castrillo, G., Werner, E., and Benincasa, F.: The WMO SDS-WAS Regional Center for Northern Africa, Middle East and Europe, E3S Web Conf., 99, 04008, https://doi.org/10.1051/e3sconf/20199904008, 2019.

Freudenthaler, V.: About the effects of polarising optics on lidar signals and the Δ90 calibration, Atmos. Meas. Tech., 9, 4181–4255, https://doi.org/10.5194/amt-9-4181-2016, 2016.

Freudenthaler, V., Esselborn, M., Wiegner, M., Heese, B., Tesche, M., Ansmann, A., MüLLER, D., Althausen, D., Wirth, M., Fix, A., Ehret, G., Knippertz, P., Toledano, C., Gasteiger, J., Garhammer, M., and Seefeldner, M.: Depolarization ratio profiling at several wavelengths in pure Saharan dust during SAMUM 2006, Tellus B: Chemical and Physical Meteorology, 61, 165–179, https://doi.org/10.1111/j.1600-0889.2008.00396.x, 2009.

Haarig, M., Ansmann, A., Engelmann, R., Baars, H., Toledano, C., Torres, B., Althausen, D., Radenz, M., and Wandinger, U.: First triple-wavelength lidar observations of depolarization and extinction-to-backscatter ratios of Saharan dust, Atmos. Chem. Phys., 22, 355–369, https://doi.org/10.5194/acp-22-355-2022, 2022.

Holben, B.N., T. Eck, I. Slutsker, A. Smirnov, A. Sinyuk, J. Schafer, D. Giles, O. Dubovik, AERONET Version 2.0 quality assurance criteria, SPIE, Goa, India, 13-17 November, 2006.

Smirnov, A., Holben, B. N., Eck, T. F., Dubovik, O., and Slutsker, I.: Cloud-Screening and Quality Control Algorithms for the AERONET Database, Remote Sensing of Environment, 73, 337–349, https://doi.org/10.1016/S0034-4257(00)00109-7, 2000.

Stein, A. F., Draxler, R. R., Rolph, G. D., Stunder, B. J. B., Cohen, M. D., and Ngan, F.: NOAA's HYSPLIT Atmospheric Transport and Dispersion Modeling System, Bulletin of the American Meteorological Society, 96, 2059–2077, https://doi.org/10.1175/BAMS-D-14-00110.1, 2015.

Velazquez Garcia, A.: Chemical and optical properties of particulate pollution in the Lille area, Northern France based on ATOLL observations, Lille University, 2023.

Velazquez-Garcia, A., Crumeyrolle, S., De Brito, J. F., Tison, E., Bourrianne, E., Chiapello, I., and Riffault, V.: Deriving composition-dependent aerosol absorption, scattering and extinction mass efficiencies from multi-annual high time resolution observations in Northern France, Atmospheric Environment, 298, 119613, https://doi.org/10.1016/j.atmosenv.2023.119613, 2023.